# Learning to Perceive the World Through Control: Empowerment-Based Representation Learning

**Mahsa Bastankhah** [1]  **Sophie Broderick** [1]  **Benjamin Eysenbach** [1]

## Abstract

In many practical reinforcement learning (RL) environments, observations are far higher-dimensional than the variables that matter for control. In this work, we ask: can we learn representations that capture only control-relevant features of the environment without relying on any reward function? We study this question through the *empowerment* objective, which maximizes an agent's influence over the environment and is widely used for unsupervised skill learning. We show that empowerment agents optimized by variational lower bound induce two distinct representations — forward and backward — that capture complementary aspects of the state, and both of which are invariant to control-irrelevant features. Thus, empowerment maximization leads agents to learn an implicit, *control-centric* model of the world. Our analysis highlights the importance of learning representations through interaction rather than from passive datasets: interaction aimed at maximizing control is essential for learning useful invariance properties, a perspective that aligns closely with the causal learning literature.

**Project website:** mahsa-bastankhah.github.io/MISL

## 1. Introduction

Learned representations have proven crucial to the success of Reinforcement Learning (RL) systems in applications from backgammon (Tesauro, 1994) to Go (Schrittwieser et al., 2020) to Grand Turismo (Wurman et al., 2022). However, Unlike vision and language models trained on passive datasets, RL agents must collect their own data through interaction, and it remains unclear what kind of interaction leads to representations that generalize to any control

task (Ilyas et al., 2019). Learning representations with theoretical gaurantees typically requires access to a reward function (e.g., bisimulation; Ferns et al. 2011; Zhang et al. 2020b) or a dataset of sufficient coverage and breadth (Du et al., 2019; Wang et al., 2022a;b; Zhang et al., 2020a). But reward-based representations may discard features that are irrelevant for the current task but essential for control in other tasks (Zhang et al., 2020b;a). And methods that rely on a fixed dataset, inherit the biases of that data (Touati & Ollivier, 2021). (Lamb et al., 2023; Efroni et al., 2022). In this paper, we instead ask: *how can an agent learn an intrinsic representation of its environment, independent of any task and any collected data?*

The causal learning literature emphasizes that agents must actively intervene in the environment to rule out spurious correlations and identify causal structures essential for generalization (Pearl, 2009; Peters et al., 2017). In this paper, we will study how empowerment, an objective predicated on interaction, is a suitable objective for driving invariant representation learning. Empowerment is the channel capacity between the policy (skill) and the resulting future outcome and it quantifies how many distinct policies can be reliably distinguished from observing their future outcomes (Gregor et al., 2016; Klyubin et al., 2005). Prior work has shown this objective drives skill learning and exploration (Gregor et al., 2016; Klyubin et al., 2005; Eysenbach et al., 2018; Zheng et al., 2025; Park et al., 2024). We will use the terms empowerment and mutual information skill learning (MISL) interchangeably in this paper.

To maximize empowerment, the agent must learn a sort of *world model* that is sufficient for predicting which actions lead to a certain outcome from an initial state. Prior work has referred to this type of model as an *inverse* dynamics model (Du et al., 2019; Ghosh et al., 2018). This model allows the agent to identify and discard policies that produce redundant outcomes, instead keeping policies that produce diverse outcomes (Eysenbach et al., 2018; Zheng et al., 2025; Park et al., 2024). In this work, we study the state representations implicitly learned by this world model.

Our main conceptual building block is to relate empowerment maximization to two representations, *forward* and *backward* representations. We show that these representa-

[1]Princeton University, USA. Correspondence to: Mahsa Bastankhah <mb6458@princeton.edu>.

*Proceedings of the 43rd International Conference on Machine Learning*, Seoul, South Korea. PMLR 306, 2026. Copyright 2026 by the author(s).

tions capture complementary aspects of state: the forward representation encodes what outcomes can be influenced, while the backward representation encodes how the state can be reached by different policies (Section 5.1).

Using these representations, we will study the interplay between data collection and representation learning in empowerment. We show that empowerment under certain assumptions induces policies that *(i)* ignore control-irrelevant features; *(ii)* maximize coverage of controllable outcomes; and *(iii)* are invariant to action-interface changes [1]. These properties respectively yield representations that are *(i)* invariant to control-irrelevant features; and *(ii)* preserve all control-relevant structure[2]; and *(iii)* invariant to action relabeling, respectively (Sections 5.2, 5.3 and 5.1). Our analysis highlights the central role of data collection; because empowerment policies maximize control over the environment, the representations they induce have these invariance properties.

Our work removes the common assumption in prior control-focused representation learning methods of access to either a reward function (Ferns et al., 2011; Zhang et al., 2020b) or expert datasets (Du et al., 2019; Wang et al., 2022a;b; Zhang et al., 2020a). We theoretically characterize the invariance properties of empowerment-based representations and empirically demonstrate their usefulness for reward maximization in high-dimensional environments.

In particular, our contributions are as follows:

1. We introduce MISL forward and backward representations as a conceptual tool for understanding the state aliasing of empowerment-maximizing agents (Sections 4 and 5.1).
2. We prove that MISL policies and representations admit solutions invariant to control-irrelevant features of the environment and to the change in action interface (Section 5.2, Section 5.1).
3. We characterize conditions under which MISL representations preserve all control-relevant features, and show that this yields a strictly more generalizable, reward-agnostic analogue of bisimulation (Section 5.3).
4. We provide didactic tabular experiments illustrating the forward–backward asymmetry of MISL representations and demonstrating that they capture all controllable features (Sections 6 and Appendix H.1).
5. We validate the noise invariance and control utility of MISL representations in high-dimensional and pixel-based control tasks, and show that they enable substantially higher downstream reward maximization perfor-

mance under severe noise (Section 6).

## 2. Related Works

**Invariant representation learning in reinforcement learning** A class of methods consists of model-based approaches. For example, Wang et al. (2022a) use a VAE to disentangle controllable, uncontrollable, and reward-relevant factors, while Wang et al. (2022b) learn causal dynamics via forward models and conditional independence tests. A common limitation of these methods is that they require learning a full environment model before irrelevant features can be removed, which can hinder scalability. Inverse-dynamics–based methods learn representations by predicting actions between state pairs (Lamb et al., 2023; Efroni et al., 2022). Although they guarantee invariance to uncontrollable features under sufficient coverage, they assume access to data collected by policies that are invariant to uncontrollable features. In contrast, we show that optimizing the empowerment objective naturally yields policies with these properties.

**Empowerment and mutual information skill learning** Empowerment, defined as the channel capacity between actions and future states, was introduced as a principle for adaptive behavior (Klyubin et al., 2005) and later incorporated into reinforcement learning via variational objectives that induce diverse skills (Gregor et al., 2016; Eysenbach et al., 2018; Park et al., 2024; Barreto et al., 2017; Park et al., 2022). While these methods focus on behavioral diversity, the representations induced by MISL have received limited theoretical analysis. Recent work studies properties of these representations under particular implementations and parameterizations (Reizinger et al., 2025) or their invariance to temporally uncorrelated features (Levy et al., 2025). In contrast, we provide a method-agnostic characterization of MISL representations and characterize their properties with respect to control. Prior work has empirically observed that empowerment-trained policies are invariant to noise (Hansen et al., 2021; Gregor et al., 2016; Mohamed & Jimenez Rezende, 2015). However to the best of our knowledge we provide the first formal characterization of these invariance properties.

## 3. Preliminaries

We consider infinite-horizon Markov decision processes (MDPs) with state space $\mathcal{S}$, action space $\mathcal{A}$, and transition dynamics $p(s_{t+1} \mid s_t, a_t)$ in a fully observable setting. The policy is denoted by $\pi(a \mid s)$, and $\gamma \in (0,1)$ is the discount factor. The discounted occupancy measure $d_\gamma^\pi(s \mid s_0)$ is the distribution of the state $S_T$ obtained by starting from $s_0$, following $\pi$, and sampling time $T \sim \text{Geom}(1-\gamma)$; equivalently, $d_\gamma^\pi(s \mid s_0) := (1-\gamma) \sum_{t=1}^\infty \gamma^{t-1} \Pr_\pi(S_t = s \mid S_0 = s_0)$.

---

[1] If two states differ only in the action interface (a relabeling of actions) but induce the same outcomes, then MISL learns interface-invariant skills: it takes the corresponding action in each state. Details in Theorem 5.1.

[2] Under certain assumptions specified in proposition 5.8

**Unsupervised skill learning** The *empowerment* of an initial state $s_0$ is defined as the channel capacity between a latent policy variable $Z$ (also called a skill), which indexes a family of policies $\pi : \mathcal{S} \times \mathcal{Z} \to \Delta(\mathcal{A})$, and a future state $S^+$ sampled from the discounted occupancy measure induced from $s_0$:

$$\mathcal{E}_\gamma(s_0) \;=\; \max_{p(z|s_0)} I(Z; S^+ \mid s_0), \qquad S^+ \sim d_\gamma^{\pi_z}(\cdot \mid s_0). \tag{1}$$

Here, with a slight abuse of notation, we write $\pi_z$ to denote the policy indexed by skill $z$. Some prior works instead use the $n$-step state distribution for a fixed horizon $n$ (Levy et al., 2025):

$$\mathcal{E}_n(s_0) \;=\; \max_{p(z|s_0)} I(Z; S_n \mid s_0). \tag{2}$$

Solving the empowerment optimization problem results in an optimal distribution over the skills that we denote by $p^*(z|s_0)$. Intuitively, empowerment measures the maximum number of distinguishable behaviors that can be executed from $s_0$, where behaviors are considered distinct if they induce reliably different terminal states (Mohamed & Jimenez Rezende, 2015, Appendix A).

Most prior work (Gregor et al., 2016; Achiam et al., 2018; Hansen et al., 2019; Sharma et al., 2019; Eysenbach et al., 2018; Zheng et al., 2025; Park et al., 2022) optimizes this objective via a variational lower bound (Barber & Agakov, 2004) (see Appendix A.1):

$$\max_{p(z|s_0)} I(Z; S^+ \mid s_0) \;\geq\; \max_{p(z|s_0),\, \theta} \mathbb{E}_{z, s^+} \left[ \log \frac{q_\theta(z \mid s_0, s^+)}{p(z \mid s_0)} \right], \tag{3}$$

This bound is tight when the discriminator is Bayes-optimal:

$$q_{\theta^*}(z \mid s_0, s^+) = \frac{p(z \mid s_0)\, d_\gamma^{\pi_z}(s^+ \mid s_0)}{p(s^+ \mid s_0)}. \tag{4}$$

In this work, we assume an expressive discriminator that attains the Bayes-optimal solution. In practice, direct optimization over distributions on policies is intractable. Therefore, prior methods (Eysenbach et al., 2018; Zheng et al., 2025; Park et al., 2022) fix a prior $p(z \mid s_0)$ and instead optimize a skill-conditioned policy $\pi_\omega(a \mid s, z)$, which induces an occupancy measure $d_\gamma^{\pi_\omega(\cdot|z)}(s^+ \mid s_0)$. This yields the practical objective

$$\max_{\omega, \theta} \mathbb{E}_z \mathbb{E}_{s^+ \sim d_\gamma^{\pi_w(\cdot|z)}} \left[ \log \frac{q_\theta(z \mid s_0, s^+)}{p(z \mid s_0)} \right], \tag{5}$$

In this work, we follow Eysenbach et al. (2022) and adopt the theoretically well-studied empowerment objective in Equation 1 for our analysis.

## 4. MISL Forward and Backward Representations

In this section, we characterize the representations learned by MISL. We will show that although the discriminator $q_\theta(z \mid s_0, s^+)$ is typically introduced as an auxiliary model for estimating and optimizing the mutual information objective, it can be viewed as a standalone representation-learning mechanism. In particular, the representations it learns from the initial and future states exhibit meaningful and useful structure. We consider the following factorization for the discriminator:

$$q_\theta(z \mid s_0, s^+) = q\left(z \mid \phi(s_0), \psi(s^+)\right), \tag{6}$$

where $\phi : \mathcal{S} \to \mathbb{R}^d$ and $\psi : \mathcal{S} \to \mathbb{R}^d$ define the *forward* and *backward* MISL representations, respectively. We adopt this terminology because, as shown later, $\phi(\cdot)$ clusters states by *similar future*, whereas $\psi(\cdot)$ clusters states by *similar past*. Note that this factorization doesn't preclude us from representing any posterior distribution if $\phi, \psi$ are expressive enough. We want these representations to retain exactly the information needed to predict the skill. So we define the notion of *minimal* representations:

**Definition 4.1** (Minimal representations). Let the posterior over skills be $p(z \mid s_0, s^+)$, and let $q(z \mid \phi(s_0), \psi(s^+))$ denote its estimate, where $\phi : \mathcal{S} \to \mathbb{R}^d$ and $\psi : \mathcal{S} \to \mathbb{R}^d$ are learned state representations. We call $(\phi, \psi)$ a pair of *minimal representations* if:

1. **Sufficiency:** the representations preserve all information needed to predict the skill:
$$q(z \mid \phi(s_0), \psi(s^+)) = p(z \mid s_0, s^+), \qquad \forall\, z, s_0, s^+.$$

2. **Minimality:** two states share the same representation *if and only if* they induce the same posterior over skills. That is,
$$p(z \mid s_0, s^+) = p(z \mid \hat{s}_0, s^+) \quad \forall z, s^+$$
$$\iff \quad \phi(s_0) = \phi(\hat{s}_0)$$

and symmetrically for $\psi(.)$.

Intuitively, these representations should capture exactly the parts of the state that are predictive of the skill and discard the rest. In our theoretical analysis, we assume that the discriminator is sufficiently regularized—e.g., via information bottleneck–style regularization—to induce minimal representations (Tishby et al., 1999; Alemi et al., 2017). However, appendix A.2 discusses why existing MISL methods satisfy this property even without explicit information-bottleneck regularization, and Appendix A.2 provides empirical support for this claim. Because we assume minimal representations, any part of the state that is not predictive of the skill is ignored by the representation. Formally:

**Definition 4.2** (Representation Invariance). Let the state be decomposed into $s = (x, e) \in \mathcal{X} \times$

$\mathcal{E}$. If the posterior is independent of the feature $e$, i.e., $p(z \mid (x, e), s^+) = p(z \mid (x, \hat{e}), s^+) \quad \forall z, s^+, x, e, \hat{e}$, then the MISL forward representation is invariant to $e$: $\phi(x, e) = \phi(x, \hat{e}), \forall x, e, \hat{e}$. A symmetric statement holds for the backward representation $\psi(\cdot)$.

Conversely, If two states lead to different posteriors, they must have different representations.

*Remark* 4.3. If there exist $s^+$ and $z$ such that $p(z \mid s_0, s^+) \neq p(z \mid \hat{s}_0, s^+)$, then $\phi(s_0) \neq \phi(\hat{s}_0)$. A symmetric statement holds for $\psi(\cdot)$.

## 5. Theoretical Results

We first characterize in Subsection 5.1 how MISL forward and backward representations alias states and highlighting a fundamental asymmetry between them. We then relate these representations to existing representation-learning frameworks. Next, in Subsection 5.2 we show that both MISL forward and backward representations are invariant to control-irrelevant state features. Finally, in Subsection 5.3, we characterize the conditions under which MISL representations capture all control-relevant features.

### 5.1. Aliasing in MISL representations

**MISL forward representations encode control over the future** The forward representation captures how states differ in the actions available to them and the outcomes of those actions. For example, two maze states at the same location but with different wall colors allow the same actions and transitions and thus share the same forward representation (Figure 1a, states $B$ and $C$).

**Theorem 5.1.** *Let $s_0$ and $\hat{s}_0$ be two states. Suppose that for every action $a \in \mathcal{A}(s_0)$ [3] there exists an action $\hat{a} \in \mathcal{A}(\hat{s}_0)$ such that $p(s^+ \mid s_0, a) = p(s^+ \mid \hat{s}_0, \hat{a})$, and vice versa, Then:*

1. *any policy selected by MISL induces the same action distribution in the two states:*
$$p^*(z \mid s_0) > 0 \implies \pi(a \mid s_0, z) = \pi(\hat{a} \mid \hat{s}_0, z);$$

2. *under the minimal representation assumption (Definition 4.1), the MISL forward representations coincide:*
$$\phi(s_0) = \phi(\hat{s}_0).$$

Proof in Appendix C.

**Implication.** Theorem 5.1 shows MISL policies have invariance to changes in the *action interface*. For example, consider two versions of the same game that differ only in the action interface: in $s_0$ pressing "A" makes the agent jump and pressing "B" makes it crouch, while in $\hat{s}_0$ the labels are swapped. The underlying dynamics are identical—only

the interface mapping changes—so a skill such as "jump" should transfer by simply executing the corresponding interface action.

Theorem 5.1 shows MISL has exactly this consistency: it learns skills (e.g., "jump" and "crouch") that remain the same across interface changes. This property is extremely useful for generalization as the agent doesn't need to re-learn when the interface is changed. Moreover, Theorem 5.1 shows that MISL also learns identical forward representations in these states. The forward representation ignores superficial interface differences and groups states by their controllability structure. We empirically verify this property in Appendix H.2.

**Advantage over forward representations in SR** This invariance is not guaranteed for representation learning methods such as successor representations (SR), which learn *on-policy* representations (Barreto et al., 2017). SR features depend on the behavior policy used to collect data and do not, by themselves, prescribe how that policy should be chosen. As a result, if the data-collection policy is not invariant to an interface change, the learned representation will not be invariant either. This creates a circular dependence: learning an invariant representation requires collecting data with an invariant policy. MISL breaks this circle by prescribing a policy objective that naturally yields interface-invariant optimal behavior.

**Geometric intuition** The geometric interpretation of empowerment by Eysenbach et al. (2022) shows that the set of achievable state occupancies across skills forms a convex polytope in the state simplex, and MISL assigns probability only to the vertex policies of this polytope [4]. In simple terms, MISL prefers "extreme" ways of affecting the future, not policies that blend the outcomes of other policies. Here, taking different (non-matching) actions in $s_0$ and $\hat{s}_0$ makes the skill behave like a blend of two different skills, so its future occupancy is a convex combination of theirs. So such policies will never be chosen by MISL. Since MISL policies take the same action at $s_0$ and $\hat{s}_0$, the two states induce exactly the same skill-outcome channel: for each skill $z$, $d_\gamma^{\pi_z}(s^+ \mid s_0)$ and $d_\gamma^{\pi_z}(s^+ \mid \hat{s}_0)$ are the same. Consequently, the induced posterior over skills is identical as well.

**MISL backward representations cluster states by how they are reached** While forward MISL representations characterize *what an agent can do next*, backward MISL representations characterize *how a state was reached*. Intuitively, if two states $s^+$ and $\hat{s}^+$ can be reached in indistinguishable ways—so that observing either state provides the same evidence about which policy was executed to reach there—then they should share the same backward representation.

**Proposition 5.2** (Backward aliasing)**.** *Suppose two states*

---

[3]$\mathcal{A}(s)$ is the set of actions available to the agent at state $s$.

[4]Refer to Lemma B.1

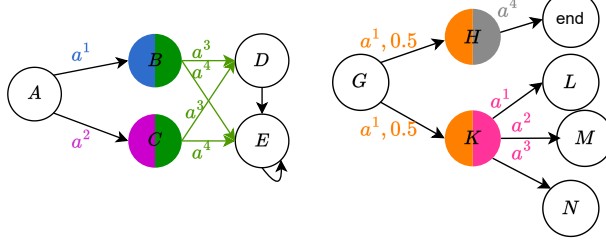

*(a) $B$ and $C$ have similar dynamics in the future hence they have the same forward representations but not the same backward representations.*

*(b) $H$ and $K$ are similarly reachable hence have the same backward representations but not the same forward representations.*

*Figure 1.* Forward and backward representations are asymmetric

$s^+$ and $\hat{s}^+$ *satisfy the following condition: for every initial state $s_0$, either (i) Both $s^+$ and $\hat{s}^+$ are unreachable from $s_0$, or (ii) There exists a constant $\alpha(s_0) > 0$ such that*

$$d_\gamma^\pi(s^+ \mid s_0) = \alpha(s_0)\, d_\gamma^\pi(\hat{s}^+ \mid s_0) \quad \forall \pi \in \Pi.$$

*Then, under the minimal representation assumption, the MISL backward representations alias these states: $\psi(s^+) = \psi(\hat{s}^+)$.*

Importantly, states that are aliased by the backward representation may have very different forward representations. For example, consider two states corresponding to the same location on a map that is next to a door; in one state the door is locked (state $H$ in Fig. 1b), while in the other it is open (state $K$ in Fig. 1b), the door is open with probability 0.5. From any initial state, both states can be reached by the same action sequences, resulting in identical backward representations. However, the open door state ($K$) allows much greater control over future outcomes than the locked-door state, and therefore has a different forward representation.

Interestingly, to the best of our knowledge, prior work has not identified this fundamental asymmetry in MISL representations and instead employs symmetric parameterizations for the forward and backward representations (Park et al., 2024; Zheng et al., 2025). In Section 6, we empirically verify this asymmetry in practice and show that enforcing symmetric representations will decrease the skill and outcome mutual information.

Backward representations are particularly well suited for representing goals. If two goal states are reached by the same policies, then from the perspective of policy training they are equivalent and should be represented similarly. This idea closely matches the notion of *actionable representations* introduced by Ghosh et al. (2018). While Ghosh et al. (2018) obtain such representations by explicitly training and comparing goal-reaching policies for every pair of states, MISL backward representations recover this structure implicitly. In Appendix D, we show that the $\ell_1$ distance

between MISL backward representations has a clear interpretation as a measure of reachability difference, making it especially useful for goal embedding.

### 5.2. Invariance to control-irrelevant features

In this section, we analyze MISL representations in MDPs whose states contain control-irrelevant components, showing that they are invariant to control-irrelevant factors.

**Definition 5.3.** Consider an MDP with state $s_t = (y_t, w_t, e_t)$ whose transition dynamics factorize as

$$p(s_{t+1} \mid s_t, a_t) = p(y_{t+1} \mid y_t, w_t, a_t)$$
$$\times\, p(w_{t+1} \mid w_t)\, p(e_{t+1} \mid e_t). \quad (7)$$

- $y_t$ is *controllable*, since its dynamics depend on the agent's action.
- $w_t$ and $e_t$ are *uncontrollable*, as their dynamics are action-independent.
- Among uncontrollable features, $w_t$ is *control-relevant* because it affects the dynamics of $y_t$, whereas $e_t$ is *control-irrelevant*.

Refer to Figure 2. For notational convenience, we group all the control-relevant features as $x_t \coloneqq (y_t, w_t)$. Throughout, when we refer to $x_t$ as the *control-relevant* features and $e_t$ as the *control-irrelevant* features. We assume a fully observable setting, so all components of $s_t$ are observed by the agent. In the main body of the paper, we analyze the factorization in Equation 7. However, all results also extend to the more general case where $w_t$ is additionally a parent of $e_{t+1}$. In that case, the transition factorization becomes

$$p(s_{t+1} \mid s_t, a_t) = p(y_{t+1} \mid y_t, w_t, a_t)$$
$$\times\, p(w_{t+1} \mid w_t)\, p(e_{t+1} \mid e_t, w_t).$$

All theoretical results still hold under a mild additional assumption. Since the analysis is more involved, we defer the full discussion and the details of the proof in this case to Appendix G.

Next, given the state factorization in Definition 5.3, we ask whether policies optimized with the empowerment objective need to attend to all components $y, w, e$. Since MISL policies maximize the diversity of reachable future outcomes, the key question is whether conditioning actions on the control-irrelevant component $e$ allows the agent to reach a broader range of future state distributions. Showing that this is not the case will allow us to prove that maximizing empowerment naturally leads to representations that are invariant to these control-irrelevant features:

**Theorem 5.4.** *Consider an MDP as in Definition 5.3.*

*There exists at least one empowerment-maximizing skill distribution*

$$p^*(z \mid s_0) \in \underset{p(z \mid s_0)}{\arg\max}\, I(Z; S^+ \mid S_0 = s_0)$$

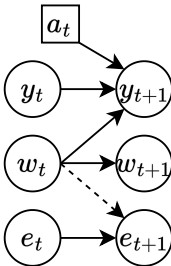

*Figure 2.* $y$ is controllable; $w$ is uncontrollable but control-relevant; $e$ is uncontrollable and control-irrelevant. If the dashed line exists, $y$ and $e$ share a confounding parent. Minimal empowerment-based representations and policies ignore $e$.

*composed of policies that are invariant to the control-irrelevant feature $e$:*

$$p^*(z \mid s_0) > 0 \implies$$
$$\pi(a \mid (x,e), z) = \pi(a \mid (x,\hat{e}), z) \quad \forall x, e, \hat{e}, a, a.$$

Proof in Appendix E.

**Proof sketch.** To obtain this result, we first show that the uncontrollable component $e^+$ cannot contribute to the mutual information objective, since it carries no information about the executed policy. As a result, empowerment depends only on the mutual information between the controllable state component $x^+$ and the skill $z$. We then show that conditioning actions on $e$ also does not help in increasing the mutual information between $z$ and $x^+$. Intuitively, reacting to $e$ cannot increase the diversity of future $x^+$ states: any policy that depends on $e$ can be matched by a (possibly non-stationary) policy that depends only on $x^+$. By Theorem 3.1 of Altman (1999), nonstationary policies do not enlarge the set of achievable discounted occupancy measures. Therefore, conditioning actions on the noise component $e$ cannot increase the mutual information. In practice, the set of $e$-invariant policies is only one subset of the optimal solutions: Theorem 5.4 does not rule out optimal policies that depend on $e$. We therefore impose a policy minimality assumption to ensure the selected optimum is noise-invariant.

**Definition 5.5** (Policy minimality). Fix a discounted occupancy kernel $d_\gamma(\cdot \mid s_0)$. Define the equivalence class

$$\Pi_d := \left\{ \pi : \ d_\gamma^\pi(s^+ \mid s_0) = d_\gamma(s^+ \mid s_0) \ \forall s_0, s^+ \right\}.$$

A *minimal policy* for $\Pi_d$ is any $\hat{\pi}_d \in \Pi_d$ that minimizes the state-action mutual information: $\hat{\pi}_d \in \arg\min_{\pi \in \Pi_d} I(S; A)$.

While theoretically we need this assumption, in practice as illustrated in our experiments in Section **??**, we observe noise invariance in MISL policies without any explicit regularization.

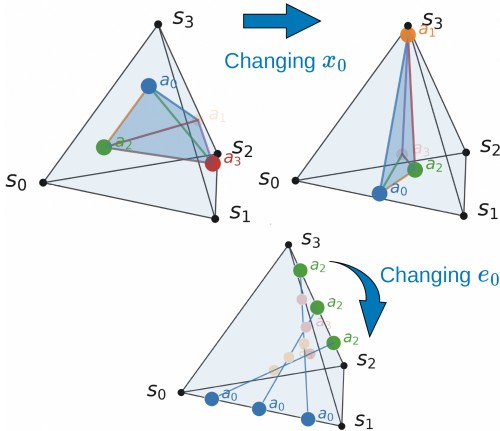

*Figure 3.* Initial states with the same $x$ but different $e$ produce similar achievable future-state distributions, differing only by rotation and translation (bottom), while states with different $x$ produce fundamentally different polytopes (top).

**Corollary 5.6.** *Consider an MDP as in Definition 5.3 with an irreducible noise process[5], under the minimal representation assumption 4.1 and the minimal policy assumption 5.5, both forward and backward MISL representations become invariant to $e_t$.*

Since according to Theorem 5.4 all minimal MISL policies choose actions independently of $e$, we can show that, given an initial state $(x_0, e_0)$ and a final state $(x^+, e^+)$, the optimal posterior used to infer which skill was executed does not need to use the noise component $e$. As a result, the posterior—and hence the learned representations—are invariant to the noise.

**Geometric intuition.** We use the geometric perspective of Eysenbach et al. (2022) to build intuition for empowerment's invariance to $e$. In this view, for a fixed start state the set of achievable next-state distributions across actions forms a *convex polytope* in the probability simplex. Empowerment selects the *extreme points* of this polytope—the actions whose induced distributions are maximally separated (saturated dots in Figure 3). Figure 3 (top) illustrates the achievable occupancy polytope in a 4-state MDP for two different start states that differ in their control-relevant feature $x$. Because these states have different controllable futures, their available next-state distributions (dots) form different polytopes, and empowerment selects different extreme actions in each case. Therefore the choice of policy and the skill posterior varies by changing $x_0$. On the other hand, Figure 3 (bottom) shows a different MDP with states $s_0 = (x,e)$, $s_1 = (x,e')$, $s_2 = (x',e)$, $s_3 = (x',e')$. Since actions can affect only $x$, the achievable next-state distri-

---

[5]Irreducible noise process is a stochastic process that for any $e^+, e$, $\sum_t \gamma^t P(E_t = e^+ \mid e_0 = e) > 0$. Or in other words, every state is reachable from another state.

butions lie on a *line segment*. Changing the initial $e$ does not alter this line in an essential way: it only shifts/rotates the same segment by moving mass between states that have a common $x$ part i.e., $s_0, s_1$ and $s_2, s_3$. Consequently, empowerment selects the same extreme actions (here $a_0$ and $a_2$) invariant of $e_0$ and the posteriors become invariant to $e_0$ as well.

### 5.3. Capturing the control-relevant state features

Representations in reinforcement learning are most useful when they discard control-irrelevant features while preserving all control-relevant ones. By capturing all control-relevant features, we mean that the representation distinguishes any two states that differ in their controllable part $y$ or in their uncontrollable but control-relevant part $w$ (Definition 5.3).

As shown in Section 5.1, forward or backward representations alone are insufficient: two states may share identical backward representations while having different forward dynamics, and vice versa. Unfortunately, in the absence of additional structural assumptions, there is no guarantee that combining forward and backward representations is sufficient to capture all control-relevant features either.

**Proposition 5.7.** *There are MDPs in which forward and backward representations together fail to capture all control-relevant features. (Proof in Appendix F.)*

However, we show that when the control-relevant dynamics i.e., $x$ dynamics are deterministic and sufficiently connected, backward representations learned using the $n$-step empowerment objective (Equation 2) alone are sufficient to capture all control-relevant features.

**Proposition 5.8.** *Consider an MDP in which the control-relevant dynamics ($x$) is deterministic and satisfies the following connectivity condition: for any two states $x^+$ and $\hat{x}^+$, there exists at least one state $x_0$ from which both $x^+$ and $\hat{x}^+$ are reachable in $n$ steps. Then the MISL backward representations learned using the $n$-step objective (Eq. 2) assign distinct representations to any two states with different $x$; that is, $\psi(x^+) \neq \psi(\hat{x}^+)$ whenever $x^+ \neq \hat{x}^+$. (Proof in Appendix F)*

**Intuition.** Intuitively, MISL learns a sufficiently rich set of skills so that all reachable control-relevant outcomes are realized by different skills (Lemma B.3). Therefore, for any two distinct future states $x^+$ and $\hat{x}^+$, there exist distinct skills $z$ and $\hat{z}$ such that executing $z$ reaches $x^+$ at time $n$, while executing $\hat{z}$ reaches $\hat{x}^+$ at time $n$. Moreover, MISL admits deterministic optimal policies (Lemma B.2).[6] As a result, the terminal state uniquely identifies the executed skill: observing $x^+$ at time $n$ implies that skill $z$ was run,

---

[6]Although multiple optimal policies may exist, the solution set always contains at least one deterministic policy.

while observing $\hat{x}^+$ implies that skill $\hat{z}$ was run. Consequently, the posterior distributions over skills differ at $x^+$ and $\hat{x}^+$, and therefore these two states have distinct backward representations.

Finally, in Appendix F we show that bisimulation with respect to reward functions that are only a function of the control-relevant part (i.e., $R((x, e)) = R(x)$) ignores control-irrelevant features as well, however, it may as well discard control-relevant features if the reward does not depend on them. As a result, bisimulation-based representations may fail to generalize to new reward functions. In contrast, MISL backward representations under the assumption of Proposition 5.8 provably capture all control-relevant features without relying on any reward signal. Consequently, MISL representations form sufficient statistics for optimizing any reward function of the form $R((x, e)) = R(x)$.

*Remark* 5.9. Under the assumptions of Proposition 5.8, MISL backward representations correspond to the most general bisimulation that preserves all control-relevant features. (Appendix F)

## 6. Empirical Validations and Experiments

In this section, we empirically validate the theoretical claims from Section 5. We answer the following research questions:

- **RQ1.** Are MISL representations invariant to control-irrelevant features?

- **RQ2.** Does this invariance make MISL representations useful for downstream tasks in high-dimensional and pixel-based environments in the presence of noise?

- **RQ3.** Do representations remain invariant to noise distributions unseen during training? (out of distribution generalization)

- **RQ4** Do MISL forward and backward representations exhibit the state aliasing and asymmetry predicted in Section 5.1? and is capturing this asymmetry necessary for estimating the empowerment properly?

- **RQ5** Do MISL backward representations capture all control-relevant features as promised by Proposition 5.8?

- **RQ6** Do the policy and discriminator networks require additional regularization to learn noise-invariant representations?

In Appendix I, we compare MISL with bisimulation (Ferns et al., 2011) and AC-state (Lamb et al., 2022). Appendix I.2 shows that bisimulation is sensitive to temporally correlated noise because its objective relies on a latent forward predictor, while MISL remains invariant to noise regardless of its temporal correlation (Figure 6). Appendix I.1 shows that AC-state representations depend on the training data: if the data is collected by a policy that even partially attends to

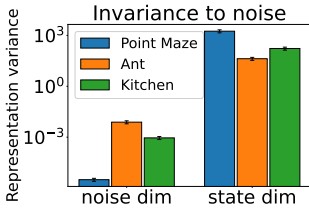

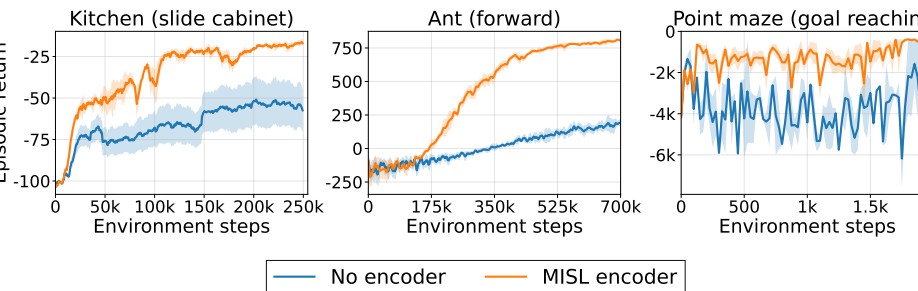

*(a)* A MISL representation of a state doesn't change by changing its noise component, and hence is noise invariant.

*(b)* MISL representations improve the downstream task performance in the presence of noise.

*Figure 4.* MISL representations are noise-invariant and useful for downstream task across environments.

noise, the learned representations are not noise-invariant. In contrast, MISL requires no pre-collected data. Furthermore, the appendix includes additional experiments on invariance under action relabeling H.2.

**Experimental setup.** We evaluate invariance to control-irrelevant features in the `point-Maze` environment based on Gym (Gymnasium Robotics, 2026), `Ant` and pixel-based Lexa `Kitchen` from METRA (Park et al., 2024). We augment the original state space (respectively 2, 29, and $64 \times 64 \times 3$ state dimensions) with additional Guassian noise dimensions (respectively 5, 10, and $10 \times 10 \times 3$ noise dimensions). These noise dimensions are uncontrollable, correlated over time, and do not affect the environment dynamics. We then study how MISL representations behave under this augmentation. We use METRA (Park et al., 2024) as the MISL algorithm[7].

**RQ1. Are MISL representations invariant to control-irrelevant features?** We examine how sensitive the learned MISL representation is to changes in individual state dimensions. For each dimension, we fix all other state variables and vary only that dimension, then record how much the learned MISL representation changes on average with varying noise versus state dimensions[8]. Figure 4a reports the results for different environments. We find that the representation variance with respect to control-relevant dimensions is approximately at least $10^3$ times larger than with respect to control-irrelevant dimensions across the three environments.

**RQ2. Does invariance improve downstream performance in the presence of noise?** To answer this question, we train a reward maximization agent[9] on top of frozen

MISL representations. The agent receives the dense environment reward. We compare the return of the agent that uses the raw noisy state versus the agent that uses the frozen MISL representations. As shown in Figures 4b, MISL representations substantially improve performance across all three environments, demonstrating their ability to ignore noise while preserving control-relevant state features.

**RQ3. Do representations generalize to out-of-distribution noise?** We evaluate the robustness of the learned representations to *unseen* noise distributions in the `Point-Maze` environment. At training time, the environment is augmented with 5-dimensional i.i.d. Gaussian noise that is temporally uncorrelated. We then test invariance under noise with temporal correlation and cross-dimensional correlation, varying the strength of correlation. As shown in Figure 6 and 15, the representations remain invariant to noise under these unseen distributions.

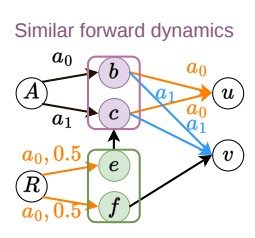

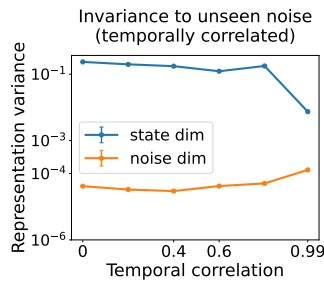

*Figure 5.* States $b$ and $c$ share forward dynamics, while $e$ and $f$ share backward dynamics, but not vice versa.

*Figure 6.* MISL representations are invariant to noise distributions unseen during training.

**RQ4. Forward and backward representation asymmetry**

To illustrate the asymmetry between forward and backward representations, we consider the simple MDP shown in Fig-

---

[7]Although METRA was originally introduced with a Wasserstein-distance objective, subsequent work has shown its equivalence to a mutual information objective (Zheng et al., 2025).

[8]Quantified by computing the trace of the covariance matrix of the representation

[9]We use Soft Actor Critic (SAC) for Point maze and Kitchen environments and PPO for the Ant environment.

ure 5. We use convex optimization tools to solve for the empowerment-maximizing distribution over skills, and the corresponding posteriors (Eq. 6). (anonymous code here). We use the average $\ell_1$ distance between posterior vectors as a proxy of representation distance. For states $b$ and $c$, we find that they have distinct backward (on average $\ell1$ norm of distance is 2) but identical forward representations[10]. Conversely, states $e$ and $f$, have identical backward but distinct forward representations (on average $\ell1$ norm of distance is 1), confirming our theory. We further ask would forcing the representations to be symmetry limit empowerment? To test this, we compare empowerment at state $A$ with and without enforcing representation symmetry. In particular, since states $b$ and $c$ have identical forward representations, we enforce identical backward representations by constraining $q(z \mid A, b) = q(z \mid A, c)$. We observe that imposing a symmetry constraint on the representations reduces the empowerment at state $A$ from 0.69 to 0.

**RQ5. Do MISL representations capture all the control-relevant state features?** To answer this question, in the `PointMaze` environment augmented with 5 Gaussian noise dimensions, we train a linear regressor $A\phi(s)$ to predict the control-relevant coordinates $(x, y)$ from the learned representations. We evaluate prediction quality using the $R^2$ metric, $R_x^2 = 1 - \frac{\sum_i (x_i - \hat{x}_i)^2}{\sum_i (x_i - \bar{x})^2}$, and similarly for $R_y^2$, where $\bar{x}$ is the mean of the $x$-coordinates. An $R^2$ value of 1 corresponds to perfect prediction. As shown in Figure 7, both $R_x^2$ and $R_y^2$ quickly approach 1, demonstrating that the learned representations fully recover the control-relevant variables despite the presence of noise. Further experiments in Appendix H.

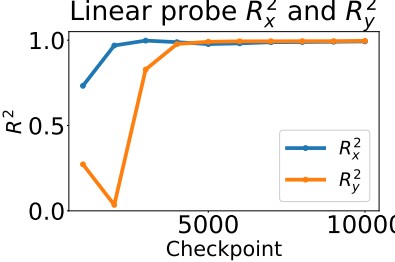

*Figure 7.* Linear probes on MISL representations can accurately recover the control-relevant features $(x, y)$ in `PointMaze`.

**RQ5. Do the networks require regularization for noise-invariance?** Theorem 5.4 requires information bottleneck regularization to guarantee noise-invariant policies and representations. However, in practice, we use the implementation of (Park et al., 2024), which applies no explicit regularization to either the policy or discriminator networks. The default implementation uses a 2-layer MLP policy with hidden width 1024 and a 2-dimensional skill vector in the `Point-Maze` environment, yet noise invariance still

---

[10]Identical representations means $\ell_1$ distance of 0

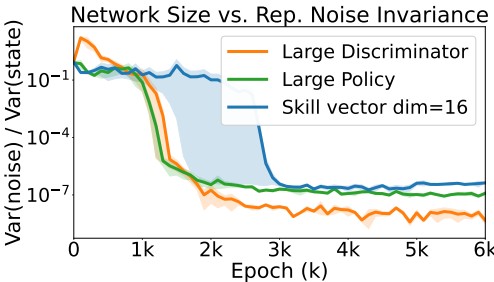

*Figure 8.* Variance of the representations to the noise relative to the state dimensions **(lower is better)**. Even as the network sizes increase, the representations remain noise-invariant without requiring explicit regularization.

emerges empirically. To investigate whether this effect is due to limited network size acting as an implicit regularizer, we scale each component independently: increasing the policy to a 5-layer MLP with width 1536, the discriminator to an 8-layer MLP with width 2048, and the skill dimension from 2 to 16. We then measure the variance of the learned representations along the 5-dimensional Gaussian noise directions versus the state dimensions. Figure 8 shows that at convergence, the representations remain consistently noise-invariant even at these larger scales, suggesting that explicit regularization may not be necessary in practice.

# 7. Conclusion

For an agent to understand the world, they should not only be able to predict the future, but also reason about which bits are uncontrollable and hence need not be predicted. Learning such a model, or the representations that support it, requires interaction. The main result of our paper is that representations learned through empowerment — a training method that requires interaction — capture only the control-relevant aspects of the environment, and nothing more. Broadly, while representation learning is often viewed as finding patterns in a dataset, or in the solution to a particular task, our analysis casts empowerment as learning to represent some minimal sufficient statistic of the world, before a task is specified.

**Limitations.** Our theoretical analysis suggests a connection between causal learning and empowerment. In reinforcement learning, causal learning has traditionally been used to identify state features that are not causally affected by actions and ignore them (Wang et al., 2022b). Our results indicate that, under certain conditions, empowerment objectives may automatically discard such irrelevant features while capturing all control-relevant ones. However, it remains unclear whether these representations are sufficient to recover the full causal structure of the control-relevant features, and if so, to what extent such a causal model would be useful. Moreover, our analysis is restricted to fully observable settings; a theoretical analysis of MISL representations in partially observable settings remains unexplored.

## Impact Statement

This paper presents work whose goal is to advance the field of Machine Learning. There are many potential societal consequences of our work, none which we feel must be specifically highlighted here.

## Acknowledgment

The authors are pleased to acknowledge that the work reported on in this paper was substantially performed using Princeton University's Research Computing resources. This material is based upon work supported by the National Science Foundation under Award No. 2441665. Any opinions, findings and conclusions or recommendations expressed in this material are those of the author(s) and do not necessarily reflect the views of the National Science Foundation. This work was supported in part by Google.

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

# A. Background Theoretical Derivations

## A.1. Empowerment variational lower bound

$$\max_{p(z|s_0)} I(Z; S_n \mid s_0)$$

$$= \max_{p(z|s_0)} \mathbb{E}_{z,s_n} \left[ \log \frac{p(z \mid s_0, s_n)}{p(z \mid s_0)} \right]$$

$$= \max_{p(z|s_0)} \mathbb{E}_{z,s_n} \left[ \log \frac{q_\theta(z \mid s_0, s_n)}{p(z \mid s_0)} \right] + \mathbb{E}_{z,s_n} \left[ \log \frac{p(z \mid s_0, s_n)}{q_\theta(z \mid s_0, s_n)} \right]$$

$$= \max_{p(z|s_0)} \mathbb{E}_{z,s_n} \left[ \log \frac{q_\theta(z \mid s_0, s_n)}{p(z \mid s_0)} \right] + \mathbb{E}_{s_n} \left[ D_{KL}(p(z \mid s_0, s_n) \| q_\theta(z \mid s_0, s_n)) \right]$$

$$\geq \max_{p(z|s_0)} \mathbb{E}_{z,s_n} \left[ \log \frac{q_\theta(z \mid s_0, s_n)}{p(z \mid s_0)} \right]$$

## A.2. The plausibility of minimal representation assumption in practice

A track of prior work on invariant representation learning uses inverse dynamics models: a representation of the state is learned and used to predict the action that caused a transition between two states. The bottleneck regularization is required to prevent the representation from capturing task-irrelevant information (Lamb et al., 2022).

In contrast, most MISL implementations learn a posterior over latent skills rather than actions. Specifically, the posterior network outputs a distribution over skills, typically parameterized as a Gaussian, and has the form $q : \mathcal{S} \times \mathcal{S} \to \Delta(\mathcal{Z})$ and usually the mean of this distribution is directly used as the state representation (current MISL methods learn just one representation for each state as opposed to distinct forward and backward)(Eysenbach et al., 2018). From Theorem 5.4 and Corollary 5.6 we know that the posterior over skill is in fact invariant to noise; therefore, since these representations are directly the skill distribution, they are invariant to noise too. In fact, as opposed to the prior work (Lamb et al., 2022) that looks at the middle layer of the neural network, hence requires information bottleneck regularization, MISL methods look at the final layer directly and therefore don't need the regularization. Moreover, since the skill distribution is highly expressive in practice, representing the states directly with the skill doesn't limit the expressivity of state representations.

# B. The Geometric Perspective on Empowerment Optimization

Eysenbach et al. (2022) provides a geometric view of the empowerment optimization problem: each achievable state distribution corresponds to a point in $\Delta^{|\mathcal{S}|-1}$, and the set of all achievable distributions forms a convex polytope.

**Lemma B.1.** *The empowerment objective assigns nonzero probability only to vertices of the polytope formed by achievable state distributions.*

*Proof.* This follows directly from Lemma 6.1 in Eysenbach et al. (2022). $\square$

**Lemma B.2.** *Under the empowerment objective, only deterministic policies are selected.*

*Proof.* Any stochastic policy induces a state distribution that is a convex combination of the distributions induced by deterministic policies that select the same actions. Consequently, stochastic policies correspond to interior points of the achievable polytope. By Lemma B.1, only vertices are selected, so stochastic policies are not chosen. $\square$

**Lemma B.3.** *For any state $s^+$, if there exists at least one policy $\pi_z$ such that $P^{\pi_z}(S_n = s^+ \mid s_0) > 0$, then under an optimal skill prior $p^*(z \mid s_0)$ we have*

$$\mathbb{E}_{z \sim p^*(z|s_0)} P^{\pi_z}(S_n = s^+ \mid s_0) > 0.$$

*Proof.* We write the empowerment objective as

$$I(Z; S_n \mid s_0) = \sum_{z, s_n} p(z \mid s_0) P^{\pi_z}(S_n = s_n \mid s_0) \log \frac{P^{\pi_z}(S_n = s_n \mid s_0)}{\sum_{\bar{z}} p(\bar{z} \mid s_0) P^{\pi_{\bar{z}}}(S_n = s_n \mid s_0)}.$$

Define the induced mixture

$$p(s_n) := \sum_{\bar{z}} p(\bar{z} \mid s_0) \, P^{\pi_{\bar{z}}}(S_n = s_n \mid s_0),$$

and form the Lagrangian (enforcing $\sum_z p(z \mid s_0) = 1$):

$$\mathcal{L} = I(Z; S_n \mid s_0) + \lambda \Big( \sum_z p(z \mid s_0) - 1 \Big).$$

Fix a particular $z$ and compute $\frac{\partial \mathcal{L}}{\partial p(z \mid s_0)}$ using the product rule. For clarity, note that

$$\log \frac{P^{\pi_{\bar{z}}}(S_n = s_n \mid s_0)}{p(s_n)} = \log P^{\pi_{\bar{z}}}(S_n = s_n \mid s_0) - \log p(s_n), \qquad \frac{\partial p(s_n)}{\partial p(z \mid s_0)} = P^{\pi_z}(S_n = s_n \mid s_0).$$

Therefore,

$$\frac{\partial \mathcal{L}}{\partial p(z \mid s_0)} = \sum_{s_n} P^{\pi_z}(S_n = s_n \mid s_0) \log \frac{P^{\pi_z}(S_n = s_n \mid s_0)}{p(s_n)}$$
$$+ \sum_{\hat{z}, s_n} p(\hat{z} \mid s_0) \, P^{\pi_{\hat{z}}}(S_n = s_n \mid s_0) \frac{\partial}{\partial p(z \mid s_0)} \Big( \log \frac{P^{\pi_{\hat{z}}}(S_n = s_n \mid s_0)}{p(s_n)} \Big) + \lambda.$$

The first line comes from differentiating the coefficient $p(z \mid s_0)$ in the $z$-summand. For the second line, only the term $-\log p(s_n)$ depends on $p(z \mid s_0)$, hence

$$\frac{\partial}{\partial p(z \mid s_0)} \Big( \log \frac{P^{\pi_{\hat{z}}}(S_n = s_n \mid s_0)}{p(s_n)} \Big) = -\frac{1}{p(s_n)} \frac{\partial p(s_n)}{\partial p(z \mid s_0)} = -\frac{P^{\pi_z}(S_n = s_n \mid s_0)}{p(s_n)}.$$

Substituting this back,

$$\frac{\partial \mathcal{L}}{\partial p(z \mid s_0)} = \sum_{s_n} P^{\pi_z}(S_n = s_n \mid s_0) \log \frac{P^{\pi_z}(S_n = s_n \mid s_0)}{p(s_n)} - \sum_{\hat{z}, s_n} p(\hat{z} \mid s_0) \, P^{\pi_{\hat{z}}}(S_n = s_n \mid s_0) \frac{P^{\pi_z}(S_n = s_n \mid s_0)}{p(s_n)} + \lambda$$

$$= \sum_{s_n} P^{\pi_z}(S_n = s_n \mid s_0) \log \frac{P^{\pi_z}(S_n = s_n \mid s_0)}{p(s_n)} - \sum_{s_n} \frac{P^{\pi_z}(S_n = s_n \mid s_0)}{p(s_n)} \Big( \sum_{\hat{z}} p(\hat{z} \mid s_0) \, P^{\pi_{\hat{z}}}(S_n = s_n \mid s_0) \Big) + \lambda$$

$$= \sum_{s_n} P^{\pi_z}(S_n = s_n \mid s_0) \log \frac{P^{\pi_z}(S_n = s_n \mid s_0)}{p(s_n)} - \sum_{s_n} P^{\pi_z}(S_n = s_n \mid s_0) + \lambda$$

$$= D_{\mathrm{KL}}\big( P^{\pi_z}(S_n = \cdot \mid s_0) \,\|\, p(\cdot) \big) - 1 + \lambda.$$

At an optimum $p^*(z \mid s_0)$, the KKT conditions imply that there exists a constant $C$ such that

$$D_{\mathrm{KL}}\big( P^{\pi_z}(S_n = \cdot \mid s_0) \,\|\, p(\cdot) \big) = C \quad \text{for all } z \text{ with } p^*(z \mid s_0) > 0,$$

and for any $z$ with $p^*(z \mid s_0) = 0$ we must have

$$D_{\mathrm{KL}}\big( P^{\pi_z}(S_n = \cdot \mid s_0) \,\|\, p(\cdot) \big) \leq C,$$

otherwise increasing $p(z \mid s_0)$ slightly would improve $\mathcal{L}$.

Now suppose for contradiction that there exists $s^+$ and a policy $\pi_{z_1}$ such that $P^{\pi_{z_1}}(S_n = s^+ \mid s_0) > 0$, but

$$\mathbb{E}_{z \sim p^*(z \mid s_0)} P^{\pi_z}(S_n = s^+ \mid s_0) = 0.$$

The latter implies $p(s^+) = 0$. Then

$$D_{\mathrm{KL}}\big( P^{\pi_{z_1}}(S_n = \cdot \mid s_0) \,\|\, p(\cdot) \big)$$

$$\geq \sum_{s \neq s^+} P^{\pi_{z_1}}(S_n = s^+ \mid s_0) \log \frac{P^{\pi_{z_1}}(S_n = s^+ \mid s_0)}{p(s^+)} + P^{\pi_{z_1}}(S_n = s^+ \mid s_0) \log \frac{P^{\pi_{z_1}}(S_n = s^+ \mid s_0)}{p(s^+)}$$

$$= \infty,$$

so $D_{\mathrm{KL}}(\cdot\|\cdot) > C$, contradicting the necessary condition that any excluded $z$ must satisfy $D_{\mathrm{KL}}(\cdot\|\cdot) \leq C$. Hence $p(s^+) > 0$ and at least one policy that reaches $s^+$ is selected by the optimal prior. $\qquad\square$

## C. Proofs of Forward and Backward State Aliasing in MISL

### Proof of Theorem 5.1

*Proof.* Consider the states $s_0$ and $\hat{s}_0$ described in the theorem statement. We show that any policy that takes *non-corresponding* actions in $s_0$ and $\hat{s}_0$ induces an occupancy measure that is a convex combination of two *interface-consistent* (symmetric) policies. Hence, by Lemma B.1, such asymmetric policies are never chosen at the empowerment optimum.

Pick two actions $a_1, a_2 \in \mathcal{A}(s_0)$ and their corresponding actions $\hat{a}_1, \hat{a}_2 \in \mathcal{A}(\hat{s}_0)$. By Lemma B.2, it suffices to consider deterministic skill-policies. Define the symmetric skill-policy for latent $z_1$:

$$\pi(a \mid s_0, z_1) = \delta(a_1), \qquad \pi(a \mid \hat{s}_0, z_1) = \delta(\hat{a}_1),$$

and the symmetric skill-policy for latent $z_2$:

$$\pi(a \mid s_0, z_2) = \delta(a_2), \qquad \pi(a \mid \hat{s}_0, z_2) = \delta(\hat{a}_2),$$

where $\delta(\cdot)$ denotes the Dirac delta. $\pi_{z_1}$ and $\pi_{z_2}$ take the same actions everywhere else.

**Truncated discounted occupancy.** For a fixed horizon $T$, define the truncated discounted occupancy under $\pi_{z_1}$ as

$$G_{\gamma,1}^T(s) := (1-\gamma) \sum_{t=1}^{T-1} \gamma^{t-1} P^{\pi_{z_1}}(S_t = s \mid S_0 = s_0),$$

and analogously under $\pi_{z_2}$,

$$G_{\gamma,2}^T(s) := (1-\gamma) \sum_{t=1}^{T-1} \gamma^{t-1} P^{\pi_{z_2}}(S_t = s \mid S_0 = s_0).$$

By the theorem assumption (action-interface relabeling), starting from $s_0$ or from $\hat{s}_0$ yields the same distribution over future trajectories under the corresponding symmetric actions. Therefore the same definitions hold if we replace the initial state $S_0 = s_0$ by $S_0 = \hat{s}_0$, i.e.,

$$G_{\gamma,i}^T(s) = (1-\gamma) \sum_{t=1}^{T-1} \gamma^{t-1} P^{\pi_{z_i}}(S_t = s \mid S_0 = \hat{s}_0), \qquad i \in \{1,2\}.$$

**Hitting-time notation.** Define the (random) first hitting times under $\pi_{z_1}$:

$$T_1 := \inf\{t \geq 1: \; S_t = s_0\}, \qquad \hat{T}_1 := \inf\{t \geq 1: \; S_t = \hat{s}_0\},$$

where the process starts from $S_0 \in \{s_0, \hat{s}_0\}$. By the theorem assumption and the fact that $\pi_{z_1}$ takes corresponding actions in $s_0$ and $\hat{s}_0$, the distribution of future trajectories (and thus of $T_1, \hat{T}_1$ and truncated occupancies) is the same whether we start from $s_0$ or from $\hat{s}_0$.

**Occupancy recursion for symmetric skills.** Expanding the discounted occupancy of $\pi_{z_1}$ starting from $s_0$ by conditioning on which of $s_0$ or $\hat{s}_0$ is hit first yields

$$d_\gamma^{\pi_{z_1}}(s \mid s_0) = \mathbb{E}\Big[\mathbf{1}\{T_1 \leq \hat{T}_1\}\big(G_{\gamma,1}^{T_1}(s) + \gamma^{T_1-1}(1-\gamma)\delta(s_0) + \gamma^{T_1} d_\gamma^{\pi_{z_1}}(s \mid s_0)\big)\Big]$$
$$+ \mathbb{E}\Big[\mathbf{1}\{T_1 > \hat{T}_1\}\big(G_{\gamma,1}^{\hat{T}_1}(s) + \gamma^{\hat{T}_1-1}(1-\gamma)\delta(\hat{s}_0) + \gamma^{\hat{T}_1} d_\gamma^{\pi_{z_1}}(s \mid \hat{s}_0)\big)\Big].$$

Since $\pi_{z_1}$ is symmetric, we have $d_\gamma^{\pi_{z_1}}(s \mid s_0) = d_\gamma^{\pi_{z_1}}(s \mid \hat{s}_0)$; denote this common quantity by $d^1(s)$. Then the recursion simplifies to

$$
\begin{aligned}
d^1(s) = \ &\mathbb{E}\Big[\mathbf{1}\{T_1 \leq \hat{T}_1\}\big(G_{\gamma,1}^{T_1}(s) + \gamma^{T_1-1}(1-\gamma)\delta(s_0) + \gamma^{T_1}d^1(s)\big)\Big] \\
&+ \mathbb{E}\Big[\mathbf{1}\{T_1 > \hat{T}_1\}\big(G_{\gamma,1}^{\hat{T}_1}(s) + \gamma^{\hat{T}_1-1}(1-\gamma)\delta(\hat{s}_0) + \gamma^{\hat{T}_1}d^1(s)\big)\Big].
\end{aligned} \tag{8}
$$

$$
d^1(s) = \frac{\mathbb{E}[G_{\gamma,1}^{\min\{T_1,\hat{T}_1\}}(s)] + \mathbb{E}_{T_1,\hat{T}_1}[\mathbf{1}\{T_1 \leq \hat{T}_1\}\gamma^{T_1-1}(1-\gamma)\delta(s_0) + \mathbf{1}\{T_1 > \hat{T}_1\}\gamma^{\hat{T}_1-1}(1-\gamma)\delta(\hat{s}_0)]}{\big(1 - \mathbb{E}[\gamma^{\min\{T_1,\hat{T}_1\}}]\big)}
$$

Similarly, we denote $d^2(s) := d_\gamma^{\pi_{z_2}}(s \mid s_0) = d_\gamma^{\pi_{z_2}}(s \mid \hat{s}_0)$ and simplify the occupancy measure:

$$
d^2(s) = \frac{\mathbb{E}[G_{\gamma,2}^{\min\{T_2,\hat{T}_2\}}(s)] + \mathbb{E}_{T_2,\hat{T}_2}[\mathbf{1}\{T_2 \leq \hat{T}_2\}\gamma^{T_2-1}(1-\gamma)\delta(s_0) + \mathbf{1}\{T_2 > \hat{T}_2\}\gamma^{\hat{T}_2-1}(1-\gamma)\delta(\hat{s}_0)]}{\big(1 - \mathbb{E}[\gamma^{\min\{T_2,\hat{T}_2\}}]\big)}
$$

Let's denote:

$$
\alpha(s) := \mathbb{E}[G_{\gamma,1}^{\min\{T_1,\hat{T}_1\}}(s)] + \mathbb{E}_{T_1,\hat{T}_1}[\mathbf{1}\{T_1 \leq \hat{T}_1\}\gamma^{T_1-1}(1-\gamma)\delta(s_0) + \mathbf{1}\{T_1 > \hat{T}_1\}\gamma^{\hat{T}_1-1}(1-\gamma)\delta(\hat{s}_0)]
$$

$$
\beta(s) := \mathbb{E}[G_{\gamma,2}^{\min\{T_2,\hat{T}_2\}}(s)] + \mathbb{E}_{T_2,\hat{T}_2}[\mathbf{1}\{T_2 \leq \hat{T}_2\}\gamma^{T_2-1}(1-\gamma)\delta(s_0) + \mathbf{1}\{T_2 > \hat{T}_2\}\gamma^{\hat{T}_2-1}(1-\gamma)(1-\gamma)\delta(\hat{s}_0)]
$$

$$
\lambda_1 := \mathbb{E}[\gamma^{\min\{T_1,\hat{T}_1\}}]
$$

$$
\lambda_2 := \mathbb{E}[\gamma^{\min\{T_2,\hat{T}_2\}}]
$$

With the new notation $d^1(s) = \frac{\alpha(s)}{\lambda_1}$ and $d^1(s) = \frac{\beta(s)}{\lambda_2}$.

Now define the *asymmetric* skill-policy $\pi_{z_{1,2}}$ by

$$
\pi(a \mid s_0, z_{1,2}) = \delta(a_1), \qquad \pi(a \mid \hat{s}_0, z_{1,2}) = \delta(\hat{a}_2),
$$

and let $\pi_{z_{1,2}}$ agree with $\pi_{z_1}$ and $\pi_{z_2}$ on every other state (i.e., the three policies differ only in their choice of action at $s_0$ and $\hat{s}_0$).

Starting from $s_0$, the trajectory repeatedly *returns* to the set $C := \{s_0, \hat{s}_0\}$. Between two consecutive returns to $C$, the policy behaves like $\pi_{z_1}$ if the return state is $s_0$ (because it will take $a_1$ next), and like $\pi_{z_2}$ if the return state is $\hat{s}_0$ (because it will take $\hat{a}_2$ next). Therefore the full discounted occupancy can be written as a discounted sum of "excursions" of two types, which yields a weighted mixture of the two symmetric occupancies.

Define rounds indexed by $i = 1, 2, \ldots$, where round $i$ starts when the process enters $C$ and ends at the next entrance to $C$. Let

$$
\tau_i := \text{the length (hitting time) of round } i \text{ until the next visit to } C, \qquad m_i := \begin{cases} 1, & \text{if round } i \text{ ends in } s_0, \\ 2, & \text{if round } i \text{ ends in } \hat{s}_0. \end{cases}
$$

We generate $(\tau_i, m_i)$ sequentially as follows.

- **Round** 1. Starting from $s_0$, sample the first hitting times $(T_1, \hat{T}_1)$ of $(s_0, \hat{s}_0)$ under the action $a_1$. If $T_1 \leq \hat{T}_1$ set $(m_1, \tau_1) = (1, T_1)$; otherwise set $(m_1, \tau_1) = (2, \hat{T}_1)$.

- **Round** $i \geq 2$. If $m_{i-1} = 1$, then the process is at $s_0$ and the policy will take $a_1$, so sample $(T_1, \hat{T}_1)$ again and set $(m_i, \tau_i)$ by the same rule as above. If $m_{i-1} = 2$, then the process is at $\hat{s}_0$ and the policy will take $\hat{a}_2$, so sample the corresponding hitting times $(T_2, \hat{T}_2)$ under $\pi_{z_2}$ and set $(m_i, \tau_i) = (1, T_2)$ if $T_2 \leq \hat{T}_2$ and $(m_i, \tau_i) = (2, \hat{T}_2)$ otherwise.

This defines a random "tree" of modes and durations $(m_i, \tau_i)_{i \geq 1}$, which keeps track of which element of $C$ is reached first at each return.

Starting from $s_0$,

$$d_\gamma^{\pi_{z_{1,2}}}(s \mid s_0) = \alpha(s) + \mathbb{E}\left[\sum_{i=1}^\infty \gamma^{\sum_{j=1}^i \tau_j}\left(\mathbf{1}\{m_i = 1\}\alpha(s) + \mathbf{1}\{m_i = 2\}\beta(s)\right)\right].$$

In particular, the occupancy under $\pi_{z_{1,2}}$ is a weighted sum of the two excursion types, and hence can be written as a convex combination of the symmetric occupancies $d^1(s)$ and $d^2(s)$ (with weights given by the discounted frequency of $m_i = 1$ versus $m_i = 2$). By Lemma B.1, empowerment (and hence MISL) can be optimized using only vertex occupancies, so such asymmetric skills receive zero probability under the optimal prior $p^*(z \mid s_0)$. The same conditioning argument applies when starting from $\hat{s}_0$, showing that $d_\gamma^{\pi_{z_{1,2}}}(\cdot \mid \hat{s}_0)$ is also a convex combination of $d^1$ and $d^2$. Therefore, only symmetric (interface-consistent) policies can have nonzero probability under MISL.

therefore we can only look at the set of symmetric policies for thoe polciies $\mathcal{Z}_{\text{symm}}$.

$$d_\gamma^{\pi_z}(s^+ \mid s_0) = d_\gamma^{\pi_z}(s^+ \mid \hat{s}_0) \qquad \forall z \in \mathcal{Z}_{\text{symm}}.$$

Therefore the empwoemrnt objective $\max_{p(z|s_0) \in \Delta^{|\mathcal{Z}_{\text{symm}}|-1}} \mathbb{E}_z \sum_{s^+} d_\gamma^{\pi_z}(s^+ \mid s_0) \log \frac{d_\gamma^{\pi_z}(s^+|s_0)}{\sum_{\bar{z}} p(\bar{z}|s_0) d_\gamma^{\pi_{\bar{z}}}(s^+|s_0)}$ is teh same across $s_0$, $\hat{s}_0$ and has the same optimal skill prior Consequently,

$$p^*(z \mid s_0) = p^*(z \mid s_0').$$

note that if the set of skills were not all symmetric skills then the occupancy of asymmetric skills starting from $s_0$ and $\hat{s}_0$ were different and we coun't donbtain this result anymore.

since both the prior and the likelihoods are symmetric, Using Bayes' rule, the corresponding optimal posteriors is also identical, for any future state $s^+$,

$$
\begin{aligned}
q^*(z \mid s_0, s^+) &= \frac{p^*(z \mid s_0)\, d_\gamma^{\pi_z}(s^+ \mid s_0)}{\sum_{\bar{z}} p^*(\bar{z} \mid s_0)\, d_\gamma^{\pi_{\bar{z}}}(s^+ \mid s_0)} \\
&= \frac{p^*(z \mid \hat{s}_0)\, d_\gamma^{\pi_z}(s^+ \mid \hat{s}_0)}{\sum_{\bar{z}} p^*(\bar{z} \mid s_0)\, d_\gamma^{\pi_{\bar{z}}}(s^+ \mid \hat{s}_0)} \\
&= q^*(z \mid \hat{s}_0, s^+),
\end{aligned}
$$

Finally, by the minimal representation assumption and Remark 4.2, equality of the Bayes-optimal posteriors implies equality of the forward representations. Hence,

$$\phi(s_0) = \phi(s_0').$$

$\square$

**Proof of Proposition 5.2**

*Proof.* Fix an initial state $s_0$ and consider the Bayes-optimal posterior over skills given a future state $s^+$:

$$q^*(z \mid s_0, s^+) = \frac{p^*(z \mid s_0)\, d_\gamma^z(s^+ \mid s_0)}{\sum_{\bar{z}} p^*(\bar{z} \mid s_0)\, d_\gamma^{\bar{z}}(s^+ \mid s_0)}.$$

If both $s^+$ and $\hat{s}^+$ are unreachable from $s_0$, then $d_\gamma^z(s^+ \mid s_0) = d_\gamma^z(\hat{s}^+ \mid s_0) = 0$ for all $z$, and the posterior is undefined on the same support for both states. However if both $s^+$ and $\hat{s}^+$ are reachable from $s_0$ and satisfy the condition that for this $s_0$ there exists a constant $\alpha(s_0) > 0$ such that

$$d^{\pi_z}(s^+ \mid s_0) = \alpha(s_0)\, d^{\pi_z}(\hat{s}^+ \mid s_0) \quad \forall z.$$

Substituting into the posterior, the factor $\alpha(s_0)$ cancels from both the numerator and denominator, yielding

$$q^*(z \mid s_0, s^+) = q^*(z \mid s_0, \hat{s}^+) \quad \forall z.$$

Thus, the two terminal states induce identical posteriors over skills for every initial state $s_0$. By the minimal representation assumption, this implies

$$\psi(s^+) = \psi(\hat{s}^+).$$

$\square$

## D. MISL Backward Representations Have Interpretable $\ell 1$ Distance

We define the reachability vector as:

$$\rho_{s_0}(s^+) := \left[ \frac{d_\gamma^\pi(s^+ \mid s_0)}{\sum_{\bar{\pi}} d_\gamma^{\bar{\pi}}(s^+ \mid s_0)} \right]_\pi. \tag{9}$$

We show that closeness of the normalized reachability vectors is equivalent to closeness of the backward representations, and vice versa.

**Proposition D.1.** *Assume the posterior $q(\cdot \mid \phi(s_0), \psi(s^+))$ denoted for simplicity by $q_{s_0}(s^+)$ is bi-Lipschitz in $\psi$, i.e., there exist $0 < \ell \leq L$ such that for all $s_0, s^+, \hat{s}^+$,*

$$\ell \|\psi(s^+) - \psi(\hat{s}^+)\|_1 \leq \|q_{s_0}(s^+) - q_{s_0}(\hat{s}^+)\|_1 \leq L \|\psi(s^+) - \psi(\hat{s}^+)\|_1,$$

*Then there exist constants $c, C > 0$ such that*

$$\frac{c}{L} \min_{s_0} \|\rho_{s_0}(s^+) - \rho_{s_0}(\hat{s}^+)\|_1 \leq \|\psi(s^+) - \psi(\hat{s}^+)\|_1 \leq \frac{C}{\ell} \max_{s_0} \|\rho_{s_0}(s^+) - \rho_{s_0}(\hat{s}^+)\|_1.$$

**Intuition.** This proposition establishes a bidirectional correspondence between reachability and backward representations. In particular, states that are reachable by similar sets of policies—i.e., that have similar reachability vectors—induce similar backward representations, and conversely. As a consequence, backward representations provide a principled notion of goal similarity. In goal-conditioned reinforcement learning, this implies that if a policy is trained to reach a target state $s^+$, then the same policy can be expected to reach another state $\hat{s}^+$ whenever $s^+$ and $\hat{s}^+$ have similar backward representations $\psi(\cdot)$, enabling generalization across the goal space.

In order to prove Proposition D.1, we first prove an auxiliary Lemma.

**Lemma D.2.** *Let $c, k, b \in \Delta^{n-1}$ with $c_{\min} := \min_i c_i$ and $c_{\max} := \max_i c_i$. Define*

$$q_1(i) := \frac{c_i k_i}{c \cdot k}, \qquad q_2(i) := \frac{c_i b_i}{c \cdot b}.$$

*Let*

$$m := \min\{c \cdot k, \ c \cdot b\}, \qquad M := \max\{c \cdot k, \ c \cdot b\}.$$

*Then the following two-sided bounds hold:*

$$\frac{1}{\left( \frac{M}{c_{min}} + \frac{M^2}{c_{min}^2} \right)} \|k - b\|_1 \ \leq \ \|q_1 - q_2\|_1 \ \leq \ \frac{2 c_{\max}}{m} \|k - b\|_1.$$

*Proof.* **Upper bound**

$$\|q_1 - q_2\|_1 = \sum_i |\frac{c_i k_i}{c.k} - \frac{c_i b_i}{c.b}|$$

$$= \sum_i |\frac{c_i k_i}{c.k} - \frac{c_i b_i}{c.b} + \frac{c_i b_i}{c.k} - \frac{c_i b_i}{c.k}|$$

$$\leq \sum_i |\frac{c_i k_i}{c.k} - \frac{c_i b_i}{c.k}| + |\frac{c_i b_i}{c.k} - \frac{c_i b_i}{c.b}|$$

$$\leq \frac{c_{\max}}{m} \|k - b\|_1 + |\frac{c.b}{c.k} - 1|$$

$$= \frac{c_{\max}}{m} \|k - b\|_1 + |\frac{c.(b - k)}{c.k}|$$

$$= \frac{c_{\max}}{m} \|k - b\|_1 + \frac{|\sum_i c_i(b_i - k_i)|}{c.k}$$

$$\leq \frac{c_{\max}}{m} \|k - b\|_1 + \frac{\sum_i |c_i(b_i - k_i)|}{c.k}$$

$$\leq \frac{c_{\max}}{m} \|k - b\|_1 + \frac{c_{\max}\|k - b\|_1}{c.k}$$

$$\leq \frac{2c_{\max}}{m} \|k - b\|_1$$

**Lower bound**

$$\|k - b\|_1 = \sum_i |\frac{c.k \ q_1(i)}{c_i} - \frac{c.b \ q_2(i)}{c_i}|$$

$$= \sum_i |\frac{c.k \ q_1(i)}{c_i} - \frac{c.b \ q_2(i)}{c_i} + \frac{c.k \ q_2(i)}{c_i} - \frac{c.k \ q_2(i)}{c_i}|$$

$$\leq \sum_i |\frac{c.k \ q_1(i)}{c_i} - \frac{c.k \ q_2(i)}{c_i}| + |\frac{c.k \ q_2(i)}{c_i} - \frac{c.b \ q_2(i)}{c_i}|$$

$$\leq \frac{M}{c_{\min}} \|q_1 - q_2\|_1 + \sum_i |\frac{q_2(i)(c.k - c.b)}{c_i}|$$

$$\leq \frac{M}{c_{\min}} \|q_1 - q_2\|_1 + \frac{1}{c_{\min}} \sum_i |q_2(i)(c.k - c.b)|$$

$$\leq \frac{M}{c_{\min}} \|q_1 - q_2\|_1 + \frac{1}{c_{\min}} \sum_i q_2(i) \times |c.k - c.b|$$

$$= \frac{M}{c_{\min}} \|q_1 - q_2\|_1 + \frac{1}{c_{\min}} (c.k)(c.b) |\frac{1}{c.k} - \frac{1}{c.b}|$$

$$= \frac{M}{c_{\min}} \|q_1 - q_2\|_1 + \frac{1}{c_{\min}} (c.k)(c.b) |\sum_i \frac{q_1(i)}{c_i} - \sum_i \frac{q_2(i)}{c_i}| \tag{1}$$

$$\leq \frac{M}{c_{\min}} \|q_1 - q_2\|_1 + \frac{1}{c_{\min}} (c.k)(c.b) \sum_i |\frac{q_1(i)}{c_i} - \frac{q_2(i)}{c_i}|$$

$$\leq \frac{M}{c_{\min}} \|q_1 - q_2\|_1 + \frac{1}{c_{\min}^2} (c.k)(c.b) \|q_1 - q_2\|_1$$

$$\leq \left(\frac{M}{c_{\min}} + \frac{M^2}{c_{\min}^2}\right) \|q_1 - q_2\|_1 \implies \|q_1 - q_2\|_1 \geq \frac{1}{\left(\frac{M}{c_{\min}} + \frac{M^2}{c_{\min}^2}\right)} \|k - b\|_1$$

Note that in (1) we used the fact that since all vectors are normalized:

$$\sum_i \frac{q_1(i)}{c_i} - \sum_i \frac{q_2(i)}{c_i} = \sum_i \frac{k_i}{c.k} - \sum_i \frac{b_i}{c.b} = \frac{1}{c.k} - \frac{1}{c.b}$$

Moreover note that $M, m$ both are expected value of $c$ therefore $c_{\min} \le m \le M \ge c_{\max}$ and $\frac{2c_{\max}}{m} \ge 2$ and $\frac{1}{\left(\frac{M}{c_{\min}} + \frac{M^2}{c_{\min}^2}\right)} \le \frac{1}{2}$ $\square$

### Proof of proposition D.1

*Proof.* Lemma D.2 shows that for any fixed initial state $s_0$, there exist constants $c, C > 0$ such that

$$c\,\|\rho_{s_0}(s^+) - \rho_{s_0}(\hat{s}^+)\|_1 \ \le \ \|q_{s_0}(s^+) - q_{s_0}(\hat{s}^+)\| \ \le \ C\,\|\rho_{s_0}(s^+) - \rho_{s_0}(\hat{s}^+)\|_1.$$

By the definition of a bi-Lipschitz posterior, taking the minimum over $s_0$ in the lower bound and the maximum over $s_0$ in the upper bound immediately yields the result of Proposition D.1. $\square$

## E. Control-Irrelevant Features Invariance results

### Proof of Theorem 5.4

*Proof.* We begin by simplifying the empowerment objective under the state factorization $s = (x, e)$ (Definition 5.3). For any policy—whether or not it conditions on $e$—the future noise component satisfies $E^+ \perp Z \mid e_0$. Consequently,

$$I(E^+; Z \mid s_0 = (x_0, e_0)) = 0.$$

This follows directly from the graphical model in Figure 9a. Although both the action selection and the skill choice $z$ may depend on $e_0$, the future noise variable $E^+$ is generated solely from $e_0$. Thus, conditioned on $e_0$, $E^+$ and $Z$ are independent.

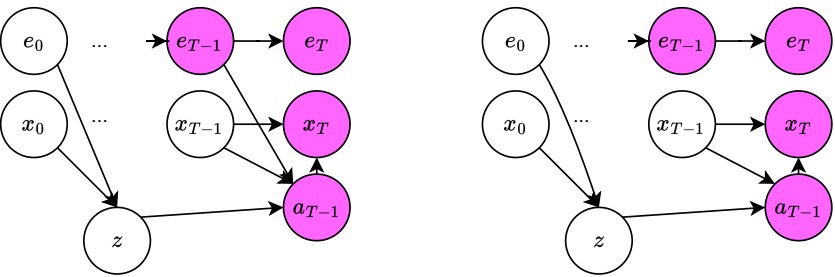

*(a)* General case, If the policy uses $e$ to make decisions, $X_T \not\perp E_T \mid s_0, z$.

*(b)* When the policy is $e$-invariant, $X_T \perp E_T \mid s_0, z$.

*Figure 9.* Graphical models illustrating the effect of policy invariance on conditional independence. $T$ is sampled from $\mathrm{Geom}(1 - \gamma)$

Therefore, we can simplify the mutual information using the chain rule:

$$I(S^+; Z \mid s_0) = I(X^+, E^+; Z \mid x_0, e_0)$$

$$= I(\underset{}{\underbrace{E^+; Z \mid x_0, e_0}_{0}}) + I(X^+; Z \mid E^+, x_0, e_0) \tag{10}$$

In general, from writing both sides of the chain rule we have:

$$I(X^+, E^+; Z \mid x_0, e_0) = I(\underset{}{\underbrace{E^+; Z \mid x_0, e_0}_{0}}) + I(X^+; Z \mid E^+, x_0, e_0)$$

$$= I(X^+; Z \mid x_0, e_0) + I(E^+; Z \mid X^+, x_0, e_0)$$

$$\overset{\mathrm{MI} \ge 0}{\Longrightarrow} I(X^+; Z \mid x_0, e_0) \le I(X^+; Z \mid E^+, x_0, e_0) \tag{11}$$

However, we show that since $e$ doesn't change the dynamics of $x$, conditioning on $E^+$ doesn't increase the MI. Or in simple words, for any skill distribution that supports $e$-variant policies, there is another skill distribution that only supports $e$-invariant policies that achieves the same mutual information.

Before proceeding with the proof, we define some notation: Recall that each skill indexes a Markovian policy. Let $\mathcal{Z}$ denote the set of all skills. We define $\mathcal{Z}_{\text{inv}}$ as the subset of skills corresponding to $e$-invariant policies (possibly nonstationary in $x$), and $\mathcal{Z}_{\text{stat}}$ as the subset corresponding to policies that are both $e$-invariant and stationary in $x$. By construction,

$$\mathcal{Z}_{\text{stat}} \subset \mathcal{Z}_{\text{inv}} \subset \mathcal{Z}.$$

1. **Step 1** We first show that for any distribution over skills $p(z \mid s_0) \in \Delta^{|\mathcal{Z}|-1}$ that assigns nonzero probability to policies that take actions based on $e$, we can construct a new distribution $p^{\text{inv}}(z \mid s_0) \in \Delta^{|\mathcal{Z}_{\text{inv}}|-1}$ whose support consists only of $e$-invariant policies (which may be nonstationary in $x$), without decreasing the mutual information. Formally,

$$I_p(X^+; Z \mid E^+, s_0) \leq I_{p^{\text{inv}}}(X^+; Z \mid x_0), \ \exists p^{\text{inv}}(z \mid s_0) \in \Delta^{|\mathcal{Z}_{\text{inv}}|-1} \tag{12}$$

where $I_p(\cdot)$ and $I_{p^{\text{inv}}}(\cdot)$ denote mutual information computed under the joint distributions induced by $p(z \mid s_0)$ and $p^{\text{inv}}(z \mid x_0)$, respectively.

2. **Step 2** In the second step of the proof, we show that restricting the skills further to non-stationary policies in $x$ does not further increase the mutual information. In particular,

$$I_{p^{\text{inv}}}(X^+; Z \mid x_0) \leq \max_{p(z|s_0) \in \Delta^{|\mathcal{Z}_{\text{inv}}|-1}} I(X^+; Z \mid x_0) = \max_{p(z|s_0) \in \Delta^{|\mathcal{Z}_{\text{stat}}|-1}} I(X^+; Z \mid x_0), \tag{13}$$

where $\mathcal{Z}_{\text{stat}}$ denotes the set of skills corresponding to $e$-invariant, Markovian, and stationary policies in $x$. This completes the proof because with step 1 together, it shows the empowerment objective in the presence of a control-irrelevant feature $e$, can be reduced to the empowerment objective only over the control-relevant part $x$ and only policies that take decisions based on $x$. To put all the results together:

$$\max_{p(z|s_0)} I(S^+; Z \mid s_0) \overset{10}{=} \max_{p(z|s_0) \in \Delta^{|\mathcal{Z}|-1}} I(X^+; Z \mid E^+, s_0)$$
$$\overset{13}{\leq} \max_{p(z|s_0) \in \Delta^{|\mathcal{Z}_{\text{stat}}|-1}} I(X^+; Z \mid x_0) \tag{14}$$

On the other hand, the other side of the inequality holds as well, i.e.,

$$\max_{p(z|s_0)} I(S^+; Z \mid s_0) \overset{10}{=} \max_{p(z|s_0) \in \Delta^{|\mathcal{Z}|-1}} I(X^+; Z \mid E^+, s_0)$$
$$\overset{11}{\geq} \max_{p(z|s_0) \in \Delta^{|\mathcal{Z}_{\text{stat}}|-1}} I(X^+; Z \mid s_0) \tag{15}$$

Therefore from Equation 14 and Equation 15:

$$\max_{p(z|s_0)} I(S^+; Z \mid s_0) = \max_{p(z|s_0) \in \Delta^{|\mathcal{Z}_{\text{stat}}|-1}} I(X^+; Z \mid x_0)$$

Moreover, we note that from Equation 10 and graphical model 9b, we note that when the policies are $e$-invariant, $X^+ \perp E^+ \mid Z, X_0$ and in that case we have:

$$I(S^+; Z \mid s_0) = I(X^+; Z \mid x_0)$$

therefore the empowerment optimization problem over the full state boils down to the optimization problem over the $x$ feature and we know that since $e$-invariant policy achieve the channel capacity according to the conclusion of step 1 , 2 above, we can instead solve the empowerment optimization problem over $x$.

Now, we provide the proof for steps 1 and 2:

**Step 1.** We expand the conditional MI:

$$I(X^+; Z \mid E^+, x_0, e_0) = \mathbb{E}_{e^+ \sim d_\gamma(e^+ \mid e_0)} I(X^+; Z \mid e^+, x_0, e_0)$$

$$= \mathbb{E}_{e^+ \sim d_\gamma(e^+ \mid e_0)} \mathbb{E}_{z \sim p(z \mid s_0), x^+ \sim d_\gamma^{\pi_z}(x^+ \mid e^+, s_0)} [\log \frac{d_\gamma^{\pi_z}(x^+ \mid e^+, s_0)}{\sum_{\bar{z}} p(\bar{z} \mid s_0) d_\gamma^{\pi_{\bar{z}}}(x^+ \mid e^+, s_0)}] \qquad (16)$$

Note that when the policy depends on $e$, $X^+$ is generally *not* conditionally independent of the future noise $E^+$:

$$X^+ \not\perp\!\!\!\perp E^+ \mid Z, s_0.$$

This follows from the causal structure in Figure 9a: if actions depend on $e$, then there exists an active path

$$E_{T-1} \rightarrow A_{T-1} \rightarrow X_T, \qquad T \sim \text{Geom}(1 - \gamma),$$

so conditioning on $(Z, s_0)$ alone does not block the dependence between $E^+$ and $X^+$.

As a result, the conditional distribution $d_\gamma^{\pi_z}(x^+ \mid s_0, z, e^+)$ generally depends on $e^+$. Intuitively, each value of $e^+$ can be viewed as defining a different communication channel between the skill $Z$ and the outcome $X^+$. If the policies (the transmission method) take actions based on the noise $e$, then these channels have different sender to receiver mutual information; however, we show that since $e$ doesn't change the dynamics of $x$, its value does not change the capacity of each channel index, meaning that no matter $e^+$ is, the capacity of the $Z, X^+$ is the same and there are $e$-invariant policies (transmission methods that ignore the noise) that achieve this capacity. We show this fact by constructing such policies.

Fix an arbitrary skill distribution $p(z \mid s_0)$, which may assign positive probability to policies that depend on $e$. Consider the mutual information conditioned on a particular realization of $e^+$,

$$I(X^+; Z \mid e^+, s_0),$$

since the policy in general could depend on $e$, the conditional MI differs based on $e$. Define the noise realization that maximizes this quantity as

$$e^* := \arg\max_{e^+} I(X^+; Z \mid e^+, s_0). \qquad (17)$$

We now construct a new distribution over skills that removes dependence on the noise process and always achieves $\max_{e^+} I(X^+; Z \mid e^+, s_0)$ no matter what the noise value is.

For each skill $z$ with policy $\pi(\cdot \mid (x, e), z)$, we construct a new skill $z'$ that indexes a policy $\pi'(\cdot \mid x, t, z')$ ($e$-invariant but non stationary) defined by

$$\pi'(a \mid x, t, z') := \mathbb{E}_{e_t \sim p(e_t \mid e^+ = e^*, e_0)}[\pi(a \mid x, e_t, z)]. \qquad (18)$$

Intuitively, $\pi'$ behaves as if the future noise realization were $e^*$, averaging over the noise trajectory consistent with that endpoint. Let $p^{\text{inv}}(z' \mid s_0)$ assign the same probability mass to $z'$ as $p(z \mid s_0)$ assigns to $z$.

Conditioned on $e^+ = e^*$, the discounted occupancy under $\pi_z$ is

$$d_\gamma^{\pi_z}(x^+ \mid e^*, s_0) = (1 - \gamma) \sum_{t=0}^{\infty} \gamma^t \Pr_{\pi_z}(X_t = x^+ \mid e^*, s_0).$$

Using the law of total probability and the fact that $e$ does not affect the dynamics of $x$, we can write

$$\Pr_{\pi_z}(X_t = x^+ \mid e^+, s_0) = \sum_{x,a} \Pr(X_{t-1} = x \mid e^*, s_0) \, \mathbb{E}_{e_{t-1} \mid e^+ = e^*, e_0}[\pi(a \mid x, e_{t-1}, z)] \, p(x^+ \mid x, a). \qquad (19)$$

By construction of $\pi'$, the expectation over $e_{t-1}$ equals $\pi'(a \mid x, t - 1, z')$, yielding

$$d_\gamma^{\pi_z}(x^+ \mid e^*, s_0) = d_\gamma^{\pi'_{z'}}(x^+ \mid s_0) = d_\gamma^{\pi'_{z'}}(x^+ \mid x_0).$$

Since the induced distribution over $X^+$ is identical,

$$I_p(X^+; Z \mid e^*, s_0) = I_{p^{\text{inv}}}(X^+; Z \mid x_0).$$

Moreover, because $e^*$ maximizes the conditional mutual information,

$$I_{p^{\text{inv}}}(X^+; Z \mid x_0) \geq I_p(X^+; Z \mid E^+, s_0).$$

This concludes the proof of step 1.

**Step 2.** Note that the new policy we constructed in step 1 is $e$-invariant but non-stationary, however in order to show the second claim of the theorem statement, i.e, the fact that $e$-invariant stationary Markovian policies are enough for optimizing the MI, we need to show that the same MI could be achieved by restricting to the stationary policies.

We directly use the result of Theorem 3.1 of Altman (1999), any discounted occupancy measure that is achievable by a (possibly non-Markovian or non-stationary) policy can also be achieved by a Markovian and stationary policy. Therefore, the set of distributions $\{d_\gamma^{\pi_z}(x^+ \mid s_0)\}$ induced by $e$-invariant Markovian policies is identical whether we restrict to stationary or nonstationary policies. This completes Step 2 of the proof.

**Proof of Corollary 5.6**

*Proof.* Theorem 5.4 shows that, under the state factorization in Definition 5.3, the empowerment objective is equivalent to

$$\max_{p(z|s_0)} I(X^+; Z \mid x_0),$$

i.e., only the control-relevant component $x$ matters for channel capacity.

By the policy minimality assumption (Definition 5.5), among all optimal solutions we select an optimal set of skills whose induced policies are $e$-invariant. In particular, for any fixed $x_0$ there exists an optimal skill prior supported only on $e$-invariant policies, and the policy minimality assumption 5.5 selects such a prior independently of the nuisance realization $e_0$. Hence the optimal prior satisfies

$$p^*(z \mid (x_0, e_0)) = p^*(z \mid (x_0, \hat{e}_0)) \qquad \forall x_0, \; e_0, \; \hat{e}_0. \tag{20}$$

Now fix any skill $z$ in the support of $p^*$ and any successor state $s^+ = (x^+, e^+)$. Since $\pi_z$ is $e$-invariant, its discounted occupancy factorizes as

$$d^{\pi_z}(s^+ \mid s_0) = d^{\pi_z}(x^+ \mid x_0) \, d_\gamma(e^+ \mid e_0),$$

where the $e$-term does not depend on $z$. Plugging this factorization into the optimal posterior gives

$$q(z \mid (x_0, e_0), s^+) = \frac{p^*(z \mid (x_0, e_0)) \, d^{\pi_z}(s^+ \mid s_0)}{\sum_{\bar{z}} p^*(\bar{z} \mid (x_0, e_0)) \, d^{\pi_{\bar{z}}}(s^+ \mid s_0)} \tag{21}$$

$$= \frac{p^*(z \mid x_0) \, d^{\pi_z}(x^+ \mid x_0) \, \cancel{d_\gamma(e^+ \mid e_0)}}{\sum_{\bar{z}} p^*(\bar{z} \mid x_0) \, d^{\pi_{\bar{z}}}(x^+ \mid x_0) \, \cancel{d_\gamma(e^+ \mid e_0)}} \tag{22}$$

$$\overset{equation\ 20}{=} q(z \mid (x_0, \hat{e}_0), s^+) \qquad \forall x_0, e_0, \hat{e}_0, z, s^+. \tag{23}$$

Therefore the posterior (and thus the forward representation induced by $q$) is invariant to the nuisance component $e_0$.

Note that since we assume the noise process is irreducible $d_\gamma(e^+ \mid e_0)$ is always positive.

The same argument applies when conditioning on the terminal state: for any $s^+ = (x^+, e^+)$ and $\hat{s}^+ = (x^+, \hat{e}^+)$, the factorization and cancellation above imply

$$q(z \mid s_0, (x^+, e^+)) = q(z \mid s_0, (x^+, \hat{e}^+)) \qquad \forall s_0, x^+, e^+, \hat{e}^+,$$

which yields $e$-invariant backward representations as well.

Importantly, note that this invariance relies on $e$-invariance of the selected optimal policies; if optimal policies were allowed to depend on $e$, the occupancy would generally not factorize and the cancellation in equation 21 would fail. $\square$

## F. Control-Relevant Features Identifiability Results

**Proof of Proposition 5.7** In general, in stochastic MDPs backward representations may cluster distinct states, since the same action can lead to different next states due to stochasticity (see Figure 1b). Unfortunately, forward representations do not necessarily resolve this ambiguity. In particular, there exist cases where states that share the same backward representation also induce identical forward representations, despite having different forward dynamics.

Consider two states $H$ and $K$ in Figure 1b with identical reachability, i.e.,

$$\forall s, a, \qquad p(H \mid s, a) = p(K \mid s, a),$$

so that $H$ and $K$ have the same backward representation. We now construct their forward dynamics so that their forward representations are also identical.

Suppose that in both $H$ and $K$ there are three available actions $a_0, a_1, a_2$, and that actions $a_1$ and $a_2$ induce identical dynamics from both states:

$$p(J \mid H, a_1) = p(J \mid K, a_1) = 1, \qquad p(L \mid H, a_2) = p(L \mid K, a_2) = 1.$$

Action $a_0$, however, induces different transitions:

$$p(J \mid H, a_0) = 0.2, \;\; p(L \mid H, a_0) = 0.8, \qquad p(J \mid K, a_0) = 0.5, \;\; p(L \mid K, a_0) = 0.5.$$

Under the MISL objective, the optimal action distributions in both states place all mass on the extreme actions:

$$p^*(a_1 \mid H) = p^*(a_2 \mid H) = \tfrac{1}{2}, \qquad p^*(a_1 \mid K) = p^*(a_2 \mid K) = \tfrac{1}{2},$$

with $p^*(a_0 \mid H) = p^*(a_0 \mid K) = 0$, since $a_0$ induces a less extreme next-state distribution.

Because only $a_1$ and $a_2$ are selected and their dynamics are identical from $H$ and $K$, the resulting posteriors coincide:

$$q(a \mid H, J) = q(a \mid K, J), \qquad q(a \mid H, L) = q(a \mid K, L).$$

Thus, $H$ and $K$ have identical forward representations as well, even though their forward dynamics differ for action $a_0$. This shows that forward and backward representations together may still fail to capture all control-relevant features in stochastic MDPs.

**Proof of Proposition 5.8** Since Theorem 5.4 and Corollary 5.6 show that control-irrelevant components are neither captured by the representation nor used by the policy, we ignore them in the analysis.

Consider two distinct states $x^+$ and $\hat{x}^+$ that are both reachable from some state $x_0$ in $n$ steps via policies $z$ and $\hat{z}$, respectively. By Lemma B.3, at least one policy that reaches $x^+$ and one policy that reaches $\hat{x}^+$ with nonzero probability must receive positive mass under the optimal empowerment distribution. Hence there are skills $z, \hat{z}$ such that,

$$p^*(z \mid x_0) > 0 \quad \text{and} \quad p^*(\hat{z} \mid x_0) > 0.$$

Because the dynamics of $x$ are deterministic, we have

$$P^{\pi_z}(X_n = x^+ \mid x_0) = 1 \quad \text{and} \quad P^{\pi_{\hat{z}}}(X_n = \hat{x}^+ \mid x_0) = 1.$$

Therefore, the backward posterior satisfies

$$q(z \mid x_0, x^+) = \frac{p^*(z \mid x_0)}{\mathbb{E}_{\bar{z}}[P^{\pi_{\bar{z}}}(X_n = x^+ \mid x_0)]} > 0,$$

while

$$q(z \mid x_0, \hat{x}^+) = \frac{p^*(z \mid x_0) \cdot 0}{\mathbb{E}_{\bar{z}}[P^{\pi_{\bar{z}}}(X_n = \hat{x}^+ \mid x_0)]} = 0.$$

Note that the denominator is well defined since $\hat{x}^+$ is reachable under some policy. An analogous argument shows

$$q(\hat{z} \mid x_0, x^+) = 0 \quad \text{and} \quad q(\hat{z} \mid x_0, \hat{x}^+) > 0.$$

Thus, $x^+$ and $\hat{x}^+$ induce different posterior distributions over policies for the same initial state $x_0$, and therefore cannot share the same backward representation.

**Discussion of Remark 5.9**  We first show that any bisimulation defined with respect to a reward function that depends only on control-relevant features, i.e., $R((x,e)) = R(x)$, must ignore the control-irrelevant component $e$. Intuitively, if two states share the same control-relevant feature $x$ but differ only in $e$, then merging them does not change either the reward or the dynamics relevant for bisimulation. As a result, any bisimulation can be made coarser by grouping together all states with the same $x$, and the coarsest bisimulation necessarily clusters states that differ only in $e$; therefore, such a bisimulation is invariant to $e$.

**Lemma F.1.** *Let $\Pi = \{C_1, \ldots, C_m\}$ be any bisimulation partition of $\mathcal{S} = \mathcal{X} \times \mathcal{E}$. Define a relation on the blocks of $\Pi$ as follows:*

$$C \sim D \quad \Longleftrightarrow \quad \exists x \in \mathcal{X} \text{ such that } (x, e) \in C \text{ and } (x, \bar{e}) \in D \text{ for some } e, \bar{e} \in \mathcal{U}.$$

*and merge all blocks that are connected under $\sim$. Denote the resulting partition by $\Pi' = \{C_1', \ldots, C_k'\}$, where $k \leq m$. Then:*

1. *$\Pi'$ is still a bisimulation.*

2. *By construction, if $(x, e) \in C_j'$, then $(x, \bar{e}) \in C_j'$ for all $\bar{e} \in \mathcal{E}$.*

*Proof.* We argue that merging blocks in this way does not violate either the reward or transition conditions of bisimulation.

**Rewards.**  Because rewards depend only on $x$, all states of the form $(x, e)$ have the same reward for any action. Within each original block of $\Pi$, rewards were already equal by the bisimulation property. Therefore, when we merge blocks that contain states with the same $x$, the reward remains constant within each new block $C_j'$.

**Transitions.**  The key observation is that every new block $C_j'$ is *complete with respect to* $e$. That is, if $(x, e) \in C_j'$, then $(x, \bar{e}) \in C_j'$ for all $\bar{e} \in \mathcal{E}$. As a result, when we compute the probability of transitioning into $C_j'$, all uncontrollable components are summed out.

Fix a state $s = (x, e)$ and an action $a$. Then the probability of transitioning into a new block $C_j'$ can be written as

$$P_{\Pi'}(C_j' \mid s, a) = \sum_{(x', e') \in C_j'} p(x', e' \mid x, e, a) = \sum_{x' \in X_j} p(x' \mid x, a),$$

where $X_j = \{x' : (x', e') \in C_j' \text{ for some } e'\}$. The sum over $e'$ disappears because each block contains *all* $e'$ for a given $x'$, and

$$\sum_{e'} p(e' \mid e) = 1.$$

Crucially, this expression depends only on $x$ and not on $e$. Therefore, any two states that share the same $x$ have identical transition probabilities into every block $C_j'$. Now Take any two states $s, t$ that were in the same original block $C_r \in \Pi$. Because $\Pi$ is a bisimulation, for every original block $C_i \in \Pi$,

$$P_{\Pi}(C_i \mid s, a) = P_{\Pi}(C_i \mid t, a) \quad \forall a.$$

Now consider any merged block $C' = \bigcup_{i \in I} C_i \in \Pi'$. Then by additivity over disjoint blocks,

$$P_{\Pi'}(C' \mid s, a) = \sum_{s' \in C'} p(s' \mid s, a) = \sum_{i \in I} \sum_{s' \in C_i} p(s' \mid s, a) = \sum_{i \in I} P_{\Pi}(C_i \mid s, a).$$

Applying the equality blockwise for $\Pi$,

$$\sum_{i \in I} P_{\Pi}(C_i \mid s, a) = \sum_{i \in I} P_{\Pi}(C_i \mid t, a) = P_{\Pi'}(C' \mid t, a).$$

So $P_{\Pi'}(C' \mid s, a) = P_{\Pi'}(C' \mid t, a)$ for all merged blocks $C' \in \Pi'$ and all $a$. Thus $\Pi'$ satisfies the transition condition and is a bisimulation. Or in simpler words, consider $t$ that was originally bisimilar to $s$ under $\Pi$. Because $\Pi$ was a bisimulation, $s$ and $t$ had the same transition probabilities into each original block. Since each new block $C_j'$ is just a union of original blocks, the transition probability into $C_j'$ is simply the sum of the probabilities into those original blocks. Thus, $s$ and $t$ still have identical transition probabilities into every block of $\Pi'$.

$\square$

## G. Theoretical Results for Coupled $e$ and $y$ Dynamics Through $w$

Here we discuss how our theoretical results extend to the more general state factorization

$$p(s_{t+1} \mid s_t, a_t) = p(y_{t+1} \mid y_t, w_t, a_t)$$
$$\times p(w_{t+1} \mid w_t)\, p(e_{t+1} \mid e_t, w_t).$$

In this setting, we require the following additional assumption.

**Assumption G.1.** We assume $W_\tau \perp E_t, Y_t \mid W_t$ and $E_\tau \perp W_t \mid E_t$ for any $\tau < t$.

This assumption means both $E$ and $W$ are more informative about their own history than the other variables about them. While this assumption might sound restrictive but it makes sense to assume that a variable effects its own future more than the future of others therefore given its own future the past variables become independent of the other variables.

**Corollary G.2.** *From assumption G.1, we can immediately infer $Y_t \perp E_t \mid W_t, A_{t-1}$ for any policy.*

Note that in general $Y_t \perp E_t \mid W_t, A_{t-1}$, since $W_{t-1}$ is a common parent of both $E_t$ and $Y_t$ (Figure 10c). However, if $W_t$ contains all the information from the history of itself that is relevant to $Y_t$, then conditioning on $E_t$ provides no additional information about $Y_t$. Intuitively, the assumption requires that $E_t$ does not reveal information about $W_{t-1}$ beyond what is already contained in $W_t$. For example, if the dynamics of $W$ are deterministic and invertible, then $W_{t-1}$ can be recovered from $W_t$, and the assumption holds.

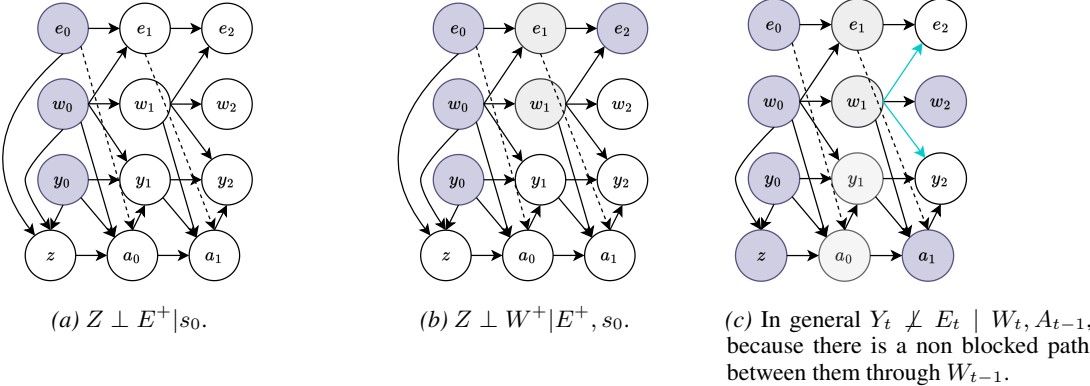

(a) $Z \perp E^+ \mid s_0$.     (b) $Z \perp W^+ \mid E^+, s_0$.     (c) In general $Y_t \not\perp E_t \mid W_t, A_{t-1}$, because there is a non blocked path between them through $W_{t-1}$.

*Figure 10.* Graphical model in the presence of a $w$ that is a confounding factor for both $e$ and $y$. Purple and gray shades are, respectively, primary and secondary shades in the Bayesian ball algorithm for determining d-separation.

$\square$

### G.1. Extending the proof of Theorem 5.4

We start by using the chain rule to rewrite the MI objective.

$$I(S^+; Z \mid s_0) = I(Y^+, W^+, E^+; Z \mid y_0, w_0, e_0)$$

$$= I(E^+; Z \mid y_0, w_0, e_0) + I(W^+; Z \mid E^+, y_0, w_0, e_0) + I(Y^+; Z \mid E^+, W^+, y_0, w_0, e_0) \quad (24)$$

The first two mutual information terms are zero. This can be verified directly from the graphical model by applying the Bayes-ball algorithm to check the corresponding d-separation relations; see respectively Figure 10a and Figure 10b.

The third term, however in general cannot be reduced to $I(Y^+; Z \mid W^+, s_0)$ even if the policy is $e$-invariant, because $Y_t \not\perp E_t \mid W_t, s_0, Z$ even if the policy is $e$-invariant due to the unblocked path through $W_{t-1}$ (Refer to Figure 10c). However under assumption G.1 and if the policy is $e$-invariant then:

Assumption G.1 and e-invariant policy $\implies I(Y^+; Z \mid E^+, W^+, y_0, w_0, e_0) = I(Y^+; Z \mid W^+, y_0, w_0)$

$$\underset{I(W^+; Z \mid E^+, y_0, w_0, e_0)=0}{\implies} I(Y^+, W^+; Z \mid E^+, s_0) = I(Y^+, W^+; Z \mid y_0, w_0)$$

Again remember the notation $X = (Y, W)$. So far we obtained that $I(S^+; Z \mid s_0) = I(X^+; Z \mid x_0)$ if the policy is $e$-invariant, but now we need to show that limiting ourselves to the set of $e$-invariant policies does not reduce the MI.

It suffices to show that for any distribution $p(z \mid s_0)$, there is another distribution of skills $p'(z' \mid x_0)$ that only has a support on $e$-invariant policies and $I(S^+; Z \mid s_0) \le I(X^+; Z' \mid x_0)$.

$$
\begin{aligned}
I(S^+; Z \mid s_0) &\overset{24}{=} I(Y^+; Z \mid E^+, W^+, s_0) \\
&= \mathbb{E}_{e^+ \sim d_\gamma(e^+ \mid e_0, w_0)}[I(Y^+; Z \mid W^+, e^+, s_0)] \\
&= \mathbb{E}_{e^+ \sim d_\gamma(e^+ \mid e_0, w_0)} \mathbb{E}_{w^+ \sim d_\gamma(w^+ \mid w_0)} \mathbb{E}_{z \sim p(z \mid s_0), y^+ \sim d_\gamma^{\pi_z}(y^+ \mid e^+, w^+, s_0)} [\log \frac{d_\gamma^{\pi_z}(y^+ \mid e^+, w^+, s_0)}{\sum_{\bar{z}} p(\bar{z} \mid s_0) d_\gamma^{\pi_{\bar{z}}}(y^+ \mid e^+, w^+, s_0)}]
\end{aligned}
$$
(25)

Similar to the proof outlined in E for any policy $\pi$ that attends to the noise $e$ we construct a new policy $\pi'$ that treats $e$ as an inner randomness so the policy itself doesn't need to attend to $e$ in order to make decisions. For each $w_0, e_0$ there is a $e^+$ that maximizes $I(Y^+; Z \mid W^+, e^+, s_0)$, let's denote that best $e^+$ by $e^*$. We use conditional independence in the markov chain graphical model and the assumption G.1 to simplify $p(y_t \mid e^+ = e^*, w^+, s_0, z)$ for any $t > 0$.

$$
\begin{aligned}
p(y_t \mid e^+ = e^*, w^+, s_0, z) = \sum_{w_{t-1}, a_{t-1}, y_{t-1}} & p(y_{t-1} \mid e^*, w^+, s_0, z)\, p(w_{t-1} \mid \cancel{e}, y_{t-1}, w^+, s_0, \cancel{z}) \\
& \times p(a_{t-1} \mid w_{t-1}, y_{t-1}, e^*, \cancel{w^+}, s_0, z)\, p(y_t \mid w_{t-1}, a_{t-1}, y_{t-1})
\end{aligned}
$$
(26)

Where by definition:

$$p(a_{t-1} \mid w_{t-1}, y_{t-1}, e^*, s_0, z) = \mathbb{E}_{e_{t-1} \sim p(e_{t-1} \mid e^+ = e^*, s_0)}[\pi(a_{t-1} \mid w_{t-1}, y_{t-1}, e_{t-1}, z)]$$
(27)

For any policy $\pi$ that attends to $e$, we can construct a new policy $\pi'$ as follows:

$$\pi_t'(a \mid y, w, z') := \mathbb{E}_{e_t \sim p(e_t \mid e^+ = e^*, e_0, w_0)}[\pi(a \mid y, w, e_t, z)].$$
(28)

We assess $p(y_t \mid w^+, s_0, z')$ for the new policy:

$$
\begin{aligned}
p(y_t \mid w^+, s_0, z') = \sum_{w_{t-1}, a_{t-1}, y_{t-1}} & p(y_{t-1} \mid w^+, s_0, z')\, p(w_{t-1} \mid y_{t-1}, w^+, s_0) \\
& \times \underbrace{p(a_{t-1} \mid w_{t-1}, y_{t-1}, s_0, z')}_{\pi_{t-1}'(a \mid y_{t-1}, w_{t-1}, z')}\, p(y_t \mid w_{t-1}, a_{t-1}, y_{t-1})
\end{aligned}
$$
(29)

No we use an induction argument, since the initial state is the same for both $\pi$ and $\pi'$, if $p(y_{t-1} \mid w^+, s_0, z') = p(y_{t-1} \mid w^+, s_0, e^*, z)$ then we immediately conclude $p(y_t \mid w^+, s_0, z') = p(y_t \mid w^+, s_0, e^*, z)$ from Equations 26, 27, 28 and 29. Therefore $d^{\pi_z}(y^+ \mid e^*, w^+, s_0) = d^{\pi_{z'}'}(y^+ \mid w^+, s_0)$ Therefore by replacing all $e$-dependent policies that have a non zero support under $p(z \mid s_0)$ with their corresponding $e$-invariant policy we will get the following:

$$
\begin{aligned}
I(Y^+; Z' \mid W^+, s_0) &= \mathbb{E}_{w^+ \sim d_\gamma(w^+ \mid w_0)} \mathbb{E}_{z' \sim p(z' \mid s_0)} \mathbb{E}_{y^+ \sim d_\gamma^{\pi_{z'}'}(y^+ \mid w^+, s_0)} \left[\log \frac{d_\gamma^{\pi_{z'}'}(y^+ \mid w^+, s_0)}{\sum_{\bar{z}} p(\bar{z} \mid s_0) d_\gamma^{\pi_{\bar{z}}'}(y^+ \mid w^+, s_0)}\right] \\
&= I(Y^+; Z \mid W^+, e^*, s_0) \\
&\ge I(Y^+; Z \mid W^+, E^*, s_0).
\end{aligned}
$$

Finally we note that, i) the construction of $\pi'$ depends on $e_0, e^*$ but for each $w_0$ there is a $e_0$ that $e^*$ that yields the maximum MI and we consider those as the inner parameters of the policy. The important fact is that the policy itself doesn't need to actively sense $e_t$ instead it is parameterized by these parameters and it means that if we search in the space of the $e$-invariant policies, the optimal policy is in that set. And ii) Similar to the argument discussed in E, Theorem 3.1 of Altman (1999) shows that the set of stationary Markovian policies is sufficient to produce any possible discounted occupancy measure therefore, although $\pi'$ is non-stationary by construction, there is a stationary policy (stationary in $w, y$) that produces the same discounted occupancy measure.

### G.2. Extending the proof of Corollary 5.6 to the coupled $y$ and $e$ through $w$

We write the Bayes optimal discriminator:

$$
\begin{aligned}
q(z \mid (y_0, w_0, e_0), s^+) &= \frac{p^*(z \mid s_0)\, d^{\pi_z}(s^+ \mid s_0)}{\sum_{\bar{z}} p^*(\bar{z} \mid s_0)\, d^{\pi_{\bar{z}}}(s^+ \mid s_0)} \\[6pt]
&= \frac{p^*(z \mid (y_0, w_0))\, \cancel{d_\gamma(w^+ \mid w_0)}\, d^{\pi_z}(y^+ \mid (y_0, w_0), w^+)\, d_\gamma^{\pi_z}(e^+ \mid (e_0, w_0, y_0), w^+, y^+)}{\sum_{\bar{z}} p^*(\bar{z} \mid (y_0, w_0))\, \cancel{d_\gamma(w^+ \mid w_0)}\, d^{\pi_{\bar{z}}}(y^+ \mid (y_0, w_0), w^+)\, d_\gamma^{\pi_{\bar{z}}}(e^+ \mid (e_0, w_0, y_0), w^+, y^+)} \quad (30) \\[6pt]
&\overset{W_{t-1} \perp Y_t \mid W_t}{=} \frac{p^*(z \mid (y_0, w_0))\, d^{\pi_z}(y^+ \mid (y_0, w_0), w^+)\, \cancel{d_\gamma(e^+ \mid (e_0, w_0), w^+)}}{\sum_{\bar{z}} p^*(\bar{z} \mid (y_0, w_0))\, d^{\pi_{\bar{z}}}(y^+ \mid (y_0, w_0), w^+)\, \cancel{d_\gamma(e^+ \mid (e_0, w_0), w^+)}} \\[6pt]
&= q(z \mid (y_0, w_0, \hat{e}_0), s^+) \qquad \forall y_0, w_0, e_0, \hat{e}_0, s^+, z
\end{aligned}
$$

Where Equation 30 comes from Theorem 5.4, the fact that empowerment picks only $e$-invariant policies in the presence of information bottleneck regularizer therefore $p^*(z \mid s_0)$ and $d^{\pi_z}(.)$ are invariant to $e$. And the third equality is the direct implication of Assumption G.1, the fact that $W_{t-1}$ of the past is independent of $Y_t$ given $W_t$, so conditioned on $w^+$, the old $w$s are all independent of $y^+$ and the policy therefore $e^+$ is independent of them too since the only possible dependence of $e^+$ to the policy or $y^+$ is through $w_{t-1}$.

A very similar argument can be done to show $q(z \mid s_0, (y^+, w^+, e^+)) = q(z \mid s_0, (y^+, w^+, \hat{e}^+)) \quad \forall y^+, w^+, e^+, \hat{e}^+, s_0, z$

## H. Experiments Backing The Theoretical Results

### H.1. RQ5.Identifiability of control-relevant features

Prior work shows that practical MISL algorithms, such as CSF (Zheng et al., 2025), can recover identifiable state representations (Reizinger et al., 2025), but these results rely on specific algorithmic choices, including contrastive losses and inner-product parameterizations. Here, we ask whether backward MISL representations capture all control-relevant features in principle, independent of the algorithm.

To show this, we design a didactic experiment in which we construct 50 random deterministic MDPs with 5 states that satisfy the connectivity assumption of Proposition 5.8. For each MDP, we solve the empowerment optimization at the most connected state (a state where every other state is reachable from it in two steps) and compute the resulting posterior over skills. We use the $\ell_1$ distance between posteriors as a proxy for backward representation distance and report the minimum distance across all state pairs and MDPs, which is always equal to or greater than 2, meaning that no two states have the same backward representation. The code is available here.

### H.2. Representation invariance to action relabeling

In order to verify Theorem 5.1, we augment the pointMaze environment as follows: in the beginning of each episode we pick one of room 0 or 1 with probability $\frac{1}{2}$, the id of the room will be indicated as part of the state i.e., $(x, y, \text{roomID})$. In room 0 the actions $a_x, a_y$ behave as expected i.e., shift the $x, y$ respectively by $a_x, a_y$; However in room 1, a scaling of 2x is applied to the actions while limiting the range of the actions by half (so the range of possible movement in room 0 and 1 is the same). States $(x, y, 0)$ and $(x, y, 1)$ are states with similar future outcomes but with different action labeling e.g., taking action $a_x$ in state $(x, y, 0)$ will produce the same outcome as taking action $\frac{a_x}{2}$ in state $(x, y, 1)$.

We measure the relative normalized mean squared error of $\phi(s, y, 0)$ and $\phi(x, y, 1)$ for $x, y$ sampled from the trajectories as

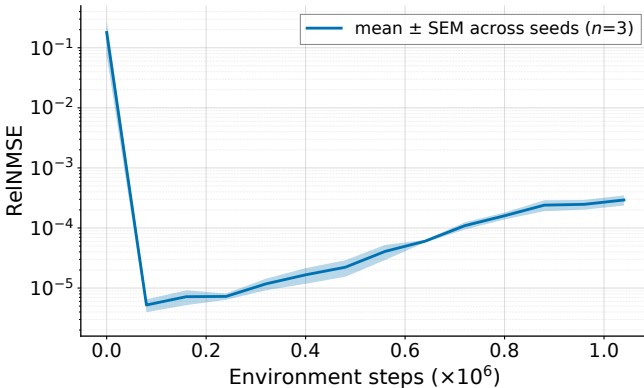

*Figure 11.* States with similar future state distribution but different action scaling have the same representation.

follows:

$$\text{RelNMSE} = \frac{\mathbb{E}_{x,y}\|\phi(x,y,0) - \phi(x,y,1)\|^2}{\mathbb{E}_{x,y}\|\phi(x,y,0)\|^2 + \mathbb{E}_{x,y}\phi(x,y,1)\|^2}$$

Lower values of RelNMSE shows that the corresponding states have a similar representations. As indicated by Figure 11, RelNMSE quickly drops and later stabilizes as training progresses, verifying our theoretical prediction in Theorem 5.1.

## I. Baseline Experiments

In this section, we compare MISL-based representation learning to prior methods—namely AC-state (Lamb et al., 2022; Efroni et al., 2022) and deep bisimulation (Zhang et al., 2020b)—that also aim to learn representations invariant to uncontrollable features. Our goal is not to provide a pure performance comparison, since these methods may perform well, or even better, in certain settings and under their intended assumptions. Instead, we focus on the conceptual limitations and practical assumptions required for them to succeed, and contrast these with empowerment-based representation learning. This emphasis aligns with the theoretical focus of the paper. We empirically challenge the key assumptions—explicit or implicit—under which these methods operate, and show where they can fail. A broader empirical comparison is left to future work.

### I.1. Multi-step inverse dynamics (AC-State)

Efroni et al. (2022); Lamb et al. (2022) present a representation-learning framework in which an encoder $f(\cdot)$ and an action predictor network $P_\theta(a_t \mid f(x_t), f(x_{t+k}); k)$ are trained jointly, where the action predictor is trained to estimate the first action $a_t$ taken to move from observation $x_t$ to a future observation $x_{t+k}$. Observations consist of a control-relevant (endogenous) part and a control-irrelevant (exogenous) part (BlockMDP setup). They show that if the training data is itself invariant to the exogenous noise, then the learned representation $f(x)$ is also noise invariant. Moreover, under additional assumptions such as bounded diameter of the endogenous dynamics and deterministic transitions, the representation provably recovers all control-relevant state features (Efroni et al., 2022; Lamb et al., 2022).

Conceptually, this framework is closely related to MISL, but there is an important difference: AC-state does not specify how the training data should be collected, yet its guarantees rely on the data already being noise invariant. In practice, this is a strong assumption. For example, the implementation of Lamb (2026) relies on pre-collected datasets for robotic manipulation and self-driving tasks. In the robotic setting, the dataset is generated using a small set of high-level actions such as "Move North," "Move South", etc. This provides structured, high-coverage, and largely noise-invariant action selection. However, in many realistic settings the environment is unknown, expert-style data is unavailable, and it is unclear how to design a noise-invariant exploration policy, especially when the noise dimensions are not known in advance.

In contrast, MISL is fully unsupervised: it does not rely on expert datasets, and it prescribes both policy learning and data collection as part of the algorithm itself. As a result, policy noise invariance emerges naturally during training rather than being assumed beforehand (Theorem 5.4).

To study the importance of noise-invariant data collection, we adapt the codebase of Lamb (2026) to a 2D point-maze

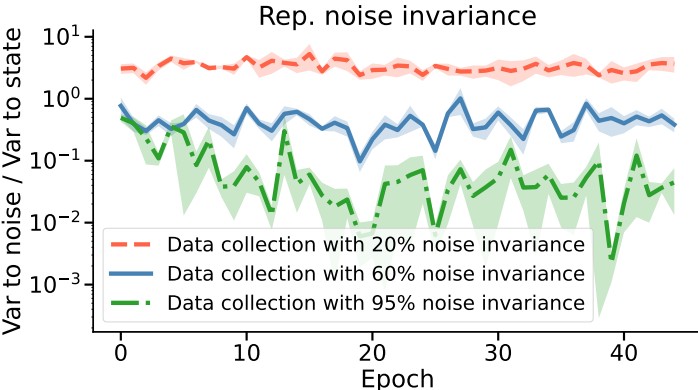

*Figure 12.* AC-state representations noise invariance (**Lower better**). This method are unable of learning noise-invariant representations if the data collection is not noise invariant.

environment augmented with one Gaussian noise dimension. We collect datasets using policies with different levels of noise dependence, and then measure the ratio of representation variance along the noise dimension to the variance along the true state dimensions. During data collection, the 2d actions are generated as a linear combination of a goal-directed component and a noise-driven component:

$$a = w_{\text{state}}\, a_{(10,10)}(x, y) + w_{\text{noise}} \tanh(n)\, \vec{u},$$

where $a_{(10,10)}(x, y)$ is the 2d action that moves toward the fixed goal at $(10, 10)$ from $(x, y)$, $n$ is the scalar noise variable and $\vec{u}$ is a fixed 2d direction. The weights $w_{\text{state}}$ and $w_{\text{noise}}$ control how noise-invariant the behavior policy is: larger $w_{\text{state}}$ and smaller $w_{\text{noise}}$ produce more noise-invariant trajectories. Figure 12 shows that when the data collection policy is highly noise invariant (e.g., $95\%$), the learned representations largely ignore the noise dimension. However, as the behavior policy becomes more noise dependent (e.g., $60\%$ or $20\%$ noise invariance), the learned representations increasingly encode the noise variable.

It is worth noting that, in tabular environments, the codebase of (Lamb, 2026) proposes a variant that jointly performs data collection and representation learning, without requiring a pre-collected dataset, and ensures noise-invariant data collection. However, this approach assumes a tabular setting with a finite set of latent states (e.g., 120 in their implementation), which limits its practicality in high-dimensional or continuous domains.

Their method builds a transition model in the learned latent space and uses Dijkstra-style planning to guide exploration. Since planning is performed entirely in the latent space, the resulting actions inherit the noise invariance of the representation. While effective in small, discrete settings, this approach does not readily scale to non-tabular environments.

### I.2. Bisimulation

Bisimulation is a widely used representation-learning principle in reinforcement learning. It aims to encode states according to how similar they are in terms of immediate reward and future outcomes. Intuitively, if two states under the current policy lead to similar rewards now and similar rewards in the future, then they should have similar representations (Ferns et al., 2011; Hansen-Estruch et al., 2022; Zhang et al., 2020b).

For example, Zhang et al. (2020b) train representations using the loss

$$J(\phi) = \left( \|z_i - z_j\|_1 - |r_i - r_j| - \gamma\, W_2\Big(\hat{P}(\cdot \mid z_i, a_i),\ \hat{P}(\cdot \mid z_j, a_j)\Big) \right)^2,$$

where $z_i = \phi(s_i)$ and $z_j = \phi(s_j)$ are the learned representations of states $s_i$ and $s_j$, and $\hat{P}(\cdot \mid z, a)$ is a learned transition model in the latent space. Here, $W_2$ denotes the 2-Wasserstein distance between the predicted next-state distributions.

This objective encourages states with similar reward and similar future behavior to be close in representation space. In practice, Zhang et al. (2020b) show that this approach works well in high-dimensional noisy environments when the reward does not depend on the noise. In such cases, states that differ only in nuisance variables but share the same endogenous state have the same reward and future reward, and are therefore mapped to similar representations.

We evaluate the noise invariance of bisimulation representations in a 2D maze augmented with 5-dimensional Gaussian noise. We use the codebase of Zhang et al. (2020b). We consider two reward functions in two different settings. The first is a standard dense reward given by the negative distance to a fixed goal at $(10, 10)$. The second is

$$-\|x - x_{\text{target}}\|,$$

which depends only on the $x$ coordinate and encourages the agent to reach any state satisfying $x = x_{\text{target}}$.

We test bisimulation under both temporally uncorrelated and temporally correlated noise. In Figure 13 (left), where the noise is temporally uncorrelated, the representation variance along the noise dimensions is much smaller than along the true state dimensions $(x, y)$, as expected. However, interestingly, when the noise is made temporally correlated (Figure 13, right), the variance of the representation along the noise dimensions increases significantly. To understand this, note that bisimulation relies on a learned transition model $\hat{P}(\cdot \mid z, a)$. When the noise is temporally correlated, the transition model can predict future noise from the current noise. As a result, even two states that differ only in their noise components induce different next-state distributions, leading to a large Wasserstein distance $W_2\left(\hat{P}(\cdot \mid z_i, a_i),\ \hat{P}(\cdot \mid z_j, a_j)\right)$, and therefore different representations, even though their endogenous (control-relevant) features are identical. This is further supported by Figure 14, which shows the variance of the mean of the transition model $\hat{P}(\cdot \mid z, a)$ (modeled as a Gaussian in this codebase) along the noise dimensions relative to the true state dimensions. The transition model clearly captures the noise, and its trend closely matches that of the learned representation in the right panel of Figure 13. This strongly suggests that the transition model is the main source of the representation's sensitivity to temporally correlated noise.

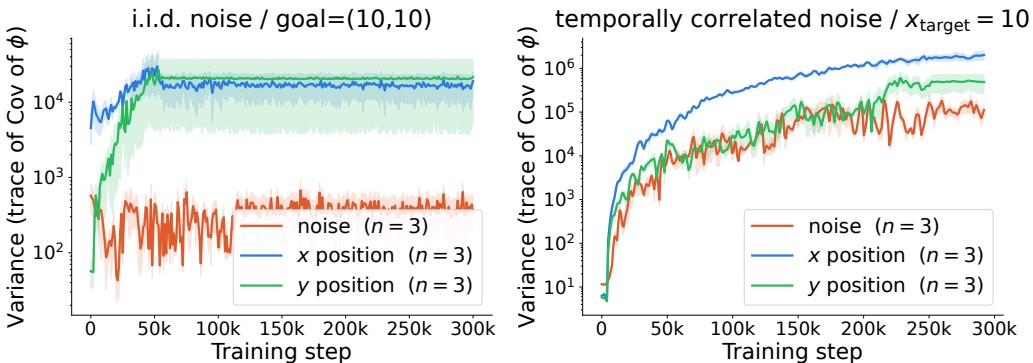

*Figure 13.* Bisimulation representation variance to noise and state dims. (lower better , higher better , higher better).

This result is surprising, since maximal bisimulation is often interpreted as enforcing noise invariance. However, in practice, methods such as Zhang et al. (2020b) do not guarantee the maximal bisimulation relation. Instead, capturing temporally correlated noise arises as a byproduct of using a predictive transition model within the learning objective.

In contrast, as shown earlier in Figure 6, temporal correlation in the noise does not break MISL; instead, it further strengthens its noise invariance, showing the conceptual superiority of MISL-based representation learning that doesn't use any predictive modeling.

The second conceptual limitation we observe in bisimulation is reward dependence. When we change the reward from the goal-based reward that depends on both $(x, y)$ (Figure 13, left) to the $x_{\text{target}}$ reward that depends only on $x$ (Figure **??**, right), the learned representation becomes much less sensitive to $y$. This highlights that bisimulation representations primarily capture the state features that the reward depends on. As a result, if the reward is not sufficiently expressive, they may fail to encode all control-relevant features.

This makes bisimulation representations well-suited for a specific known reward, but less suitable as a general pretraining objective when the downstream task is unknown or may change. In contrast, MISL-based representations aim to capture all control-relevant features, rather than only those emphasized by a particular reward. This makes them more appropriate for adaptation to new reward functions at test time.

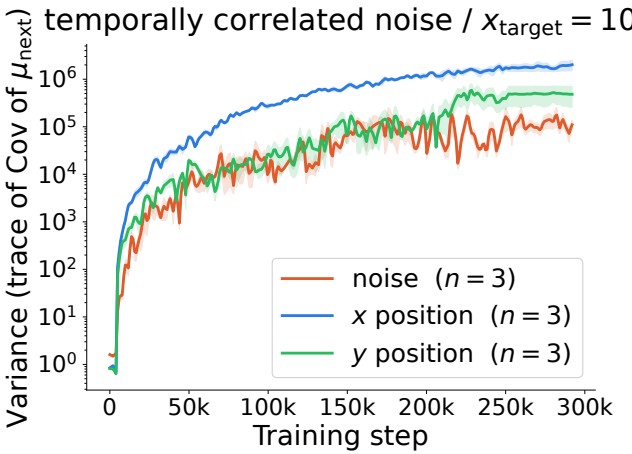

*Figure 14.* The variance of the **mean of the transition model** to noise and state dims. (lower better , higher better , higher better).

## J. Additional figures

We evaluate whether MISL representations trained with 5-dimensional i.i.d. Gaussian noise remain invariant when tested on Gaussian noise with cross-dimensional correlations in the `Point-Maze` environment. Figure 15 shows that, even as the correlation strength increases, the representations maintain low variance along the noise dimensions and high variance along the state dimensions, demonstrating strong generalization to out-of-distribution noise distributions.

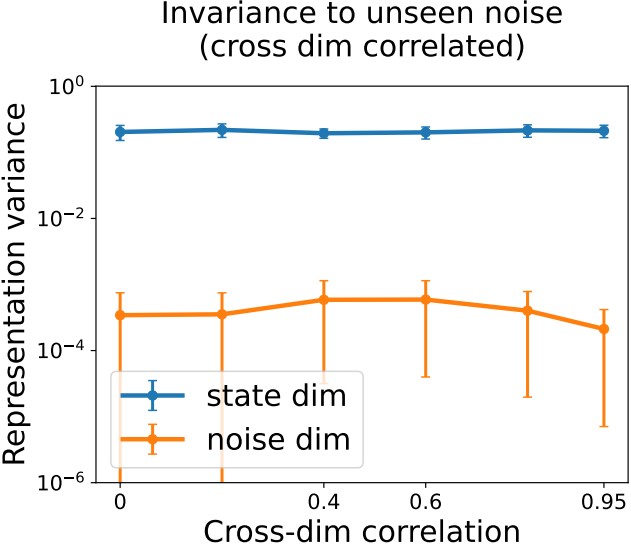

*Figure 15.* MISL representations trained with i.i.d. noise dimensions are also invariant to correlated noise dimensions.

