# OpenReview forum: "Learning to Perceive the World Through Control: Empowerment-Based Representation Learning"
_ICML.cc/2026/Conference — ICML 2026 regular_

### Official Review · Reviewer_S9NU · 2026-02-19

**Soundness:** 2
**Presentation:** 3
**Significance:** 2
**Originality:** 3
**Overall Recommendation:** 4
**Confidence:** 5

**Summary:**

This paper asks and answers a question ''can we learn representations that capture only control-relevant features of the environment?'' to show the effectiveness of ''control-centric'' representation learning in RL. Both theoretical and empirical evidence are provided in support of this claim.

**Compliance With Llm Reviewing Policy:**

Affirmed.

**Final Justification:**

All my concerns are satisfied.

**Key Questions For Authors:**

I do not have further questions. The paper is generally clear; my concerns are primarily about novelty/assumptions and the limited experimental scope.

**Limitations:**

Yes.

**Strengths And Weaknesses:**

Strength:
The paper is clearly written, and its contributions are clearly articulated and easy to understand.

Weakness:
1. The problem studied is fairly trivial. High-dimensional multimodal inputs (e.g., images) evidently contain substantial redundant information, and extracting decision-relevant features from such observations has long been a mainstream direction in RL [1]. This includes what the paper calls “control-centric feature learning,” whose effectiveness has already been demonstrated by a large body of self-supervised RL work. Even aside from prior strong results, the intuition that focusing on control-relevant factors—i.e., aspects of the observation that can be influenced by the agent’s actions—can improve policy learning efficiency is rather self-evident.
2. The core contribution relies heavily on the assumption of deterministic system dynamics, which does not hold in most real-world settings.
3. The experimental evaluation is too thin, limited to simple control tasks with low-dimensional inputs, and is insufficient to substantiate the paper’s claimed contributions.

[1] Schwarzer M, Anand A, Goel R, et al. Data-Efficient Reinforcement Learning with Self-Predictive Representations[C]//International Conference on Learning Representations, 2021.

---

> ### Author Rebuttal · Authors · 2026-03-31
>
> > has long been a mainstream direction in RL
>
> We agree that learning useful representations for decision-making has long been a mainstream direction in RL. However, in this work, we show for the first time that the **empowerment objective** can induce representations that isolate only the **control-relevant features**.
>
> The work cited by the reviewer, *Data-Efficient Reinforcement Learning with Self-Predictive Representations*, uses prediction-based models to learn temporal patterns in the environment. These methods do not capture only the control-relevant features; they capture any temporally correlated features, even if not controllable by the agent. This can be especially expensive in high-dimensional state spaces and is unhelpful for downstream tasks, since the agent cannot use those variables to do control, even if they are predictable.
>
> The novelty of our work is that we show how the empowerment objective can be used to capture **only the control-relevant features**. In particular, for the first time, we define the notions of forward and backward representations for the empowerment objective and characterize their invariance properties and other key features.
>
> There are also methods such as **AC-State** that aim to ignore control-irrelevant features by learning the inverse dynamic. However, in our AC-State baseline experiments, discussed in the response to the reviewer _y1ac_, we highlight one of their weakness compared to MISL (our method). In particular, AC-state methods require a pre-collected expert dataset, and the quality of that dataset determines the learned representation. For instance, the actions in the dataset itself should already be noise-invariant. But if we do not know the environment well in advance, it is unclear how such a dataset can be collected. Whereas, empowerment-based objectives prescribe the policy training along with the representation learning, resulting automatically in noise-invariant policies without any pre-collected datasets. In our baseline experiments, we show that with imperfect datasets, AC-state captures non-controllable features.
>
> Another conceptually related method is **bisimulation**. Bisimulation aliases states with similar reward in the future, and therefore, it is tied to a specific reward function and is not suitable for unsupervised pretraining since it will not capture the control-relevant features that don't change the reward, even if those features are useful for other control tasks. In the rebuttal response of reviewer _7AK9_, we added bisimulation baseline experiments showing that this method is susceptible to temporally correlated noise due to implicit design choices. While MISL-based methods are invariant to both temporally correlated and uncorrelated noise, in fact, they ignore anything that is irrelevant to control, no matter what its temporal properties are (Fig 5 of the paper).
>
> > The core contribution relies heavily on the assumption of deterministic system dynamics
>
> This assumption is only needed for Propositions 5.7 and 5.8, which concern the identifiability of the learned representations. These are not the most important contributions of the paper. Our main contributions are the formalization of the forward and backward representations, and the theoretical and empirical study of their invariance properties.
>
> That said, to stress-test the identifiability results beyond the deterministic setting, we conducted an additional **Identifiability in Stochastic MDPs** experiment, described in our response to reviewer _Y1AC_. The results show that, although the deterministic assumption is needed to prove some of our theoretical statements, the identifiability property still holds empirically in stochastic settings.
>
>
> > The experimental evaluation is too thin, limited to simple control tasks with low-dimensional inputs, and is insufficient to substantiate the paper’s claimed contributions.
>
> We have added **experiments in higher-dimensional settings**, including **MuJoCo Ant** and the **pixel-based Lexa Kitchen** environment; please refer to the reviewer _PLSU_'s response for details. These new results show that MISL representations remain invariant to high-dimensional noise in these environments, and that they are also useful for downstream reward maximization in MuJoCo Ant. In addition, we have included further experimental results in response to reviewer _Y1AC_ demonstrating the **invariance of the learned representations to action relabeling**. Hence, backing our theoretical results more strongly.

---

> > ### Author Rebuttal · Reviewer_S9NU · 2026-04-01
> >
> > The rebuttal meaningfully alleviates my concern that the paper was validated only on toy settings. In particular, the new Ant experiment shows that the learned MISL representation can be useful for downstream control in a modern continuous-control benchmark, which makes me comfortable raising my score to 3.
> >
> > That said, for the pixel-based Kitchen experiment, the rebuttal currently provides representation-level evidence (noise invariance) rather than downstream policy-learning results. This is helpful, but it still leaves open whether the learned representation is actually useful for control in image-based environments. If the authors were able to further show that MISL representations also improve policy learning performance on Kitchen, I would view the empirical support as substantially stronger and would likely raise my score to 4.
> > -----------
> > I have seen the latest feedback from the authors. I am impressed by the new experimental results. I raise my score to 4.

---

> > > ### Author Response · Authors · 2026-04-04
> > >
> > > Thanks to the reviewer for the comments and for raising the score.
> > >
> > > We tested MISL representations on the **slide cabinet task** in the Kitchen environment. In this task, the agent must move the robotic arm and slide open the cabinet door at the top. The reward function is dense and is based on the negative distance between the arm and the cabinet door and their target positions. We augment the
> > > 64×64×3 image observations with Gaussian noise. Specifically, we add noise in 4×4×3 blocks, where the noise is the same within each block, so the noise has spatial correlation and makes the denoising harder.
> > >
> > > When we train the agent directly from raw pixels + CNN, learning is very slow, and the behavior is noisy and unreliable. As shown in [This Video](https://anonymous.4open.science/r/MISL_representations-20ED/rl-video-episode-300-noisy.mp4), the agent can reach the cabinet, but it cannot slide it open smoothly and shows a lot of unnecessary swaying. It only manages to open the cabinet near the end of the episode.
> > >
> > > We then use our 24-dimensional MISL representation. This encoder is placed on top of the CNN and is frozen during reward-maximization training. With this encoder, the behavior improves significantly. Although the behavior is still slightly noisy, the agent is able to slide the cabinet open successfully much earlier in the episode and much more reliably [Watch The Video](https://anonymous.4open.science/r/MISL_representations-20ED/rl-video-episode-300-encoder.mp4).
> > >
> > > The training return plot can be found in [this figure](https://anonymous.4open.science/r/MISL_representations-20ED/noisy_vs_encoder_goal2.pdf).

---

### Official Review · Reviewer_y1ac · 2026-03-09

**Soundness:** 3
**Presentation:** 3
**Significance:** 3
**Originality:** 3
**Overall Recommendation:** 4
**Confidence:** 3

**Summary:**

This paper studies how representations emerge in mutual information skill learning. It shows that the Bayesian optimal discriminator can be viewed as decomposing into two complementary representations: a forward representation, which captures how the agent’s actions influence future outcomes, and a backward representation, which captures how states are reached. The authors also analyze several invariance properties of empowerment optimization, including invariance to control-independent features and to relabeling of the action interface. Experiments on tabular Markov decision processes and noisy point-maze tasks provide qualitative evidence for the theoretical claims and suggest that these representations can be useful for downstream reinforcement learning, particularly when observations are noisy.

**Compliance With Llm Reviewing Policy:**

Affirmed.

**Final Justification:**

Fully address and keep positive score

**Key Questions For Authors:**

1.	How sensitive are the empirical results to the capacity of the discriminator and the training procedure? Do you have diagnostics that indicate whether the learned representations satisfy the minimality condition described in Definition 4.1?

2.	Theorem 5.4 yields the existence of e-invariant optimal policies, but also admits non-invariant optima. In practical training, what biases (architectural, regularization, entropy terms) make the invariant solutions likely to be selected?

3.	Can you provide comparisons to PPE/AC-State or action-bisimulation on the same noisy point-maze (and, ideally, on a pixel-based distractor environment) to quantify the benefits of empowerment-driven interaction versus offline multi-step inverse/bisimulation?

4.	Proposition 5.8 assumes deterministic control-relevant dynamics. Do your point-maze dynamics satisfy determinism precisely? If you add small stochasticity to the x-dynamics, how do the backward representations degrade in practice?

5.	Your invariance metric measures representation variance to dimension-wise perturbations. Could you report complementary tests (e.g., HSIC between code and e, or a probing classifier predicting e from the codes) to strengthen the invariance claim?

6.	Does the action-interface invariance (Theorem 5.1) hold empirically when you synthetically relabel actions in a task? A simple experiment relabeling the action mapping in half the episodes would be compelling.

**Limitations:**

Yes

**Strengths And Weaknesses:**

## Soundness

The forward/backward factorization via a Bayes-optimal discriminator is a clean abstraction, and the aliasing asymmetry is well supported (Prop. 5.2). The action-interface invariance (Thm. 5.1) follows convincingly from the polytope/extreme-point view and a careful argument showing asym- metric policies induce mixed occupancies that are suboptimal under empowerment.

The noise-invariance analysis (Thm. 5.4) leverages a sensible factorization of state into control- relevant x and control-irrelevant e, and shows that conditioning on e cannot increase achievable mutual information in x. However, the need for a policy minimality selection rule (Def. 5.5) to exclude non-invariant optimal policies is a key caveat; in practice, stochastic gradient training may not implicitly prefer such solutions without explicit regularization. This reduces the force of the claim from “MISL learns invariance” to “MISL admits invariant optima.”
The identifiability-style claim (Prop. 5.8) is a helpful complement, but its determinism/connectedness assumptions limit applicability.


## Presentation

the paper is clearly written. There are a few minor issues, such as small spelling errors and formatting problems in Formula (5) and Appendix C. It would also help to provide a clearer operational definition or training diagnostic for the minimality assumption, which would make the theoretical discussion easier to connect to practical implementations.

## Significance

The paper helps clarify how interactions with the environment can lead to control-focused representations and why such representations may have desirable invariance properties. This perspective could be helpful when designing unsupervised pre-training methods for reinforcement learning, especially in settings with observation noise or changing observation interfaces.
The main contribution is conceptual clarity rather than a large empirical advance. The impact would be stronger if the paper included direct empirical comparisons with related approaches such as PPE/AC-State or action dual-simulation methods.


## Originality

The work decomposes MISL posterior probabilities into forward and backward representations and studies their invariance properties. While the intuition behind action-interface invariance is reasonable, it has not been clearly formalized in earlier MISL work.
At the same time, the paper focuses primarily on inverse or multi-step inverse approaches and does not discuss related frameworks such as EX-BMDP/PPE or AC-State, which provide theoretical guarantees for invariance and identifiability under certain assumptions.

---

> ### Author Rebuttal · Authors · 2026-03-31
>
> Thanks to the reviewer for the thoughtful and helpful comments.
>
> > AC-State or action-bisimulation
>
> For bisimulation baseline experiments, please refer to the response to reviewer 7AK9.
>
> ## AC-State Baseline Experiments
> We use the codebase of [1] and adapt it to our PointMaze environment. Our goal is not to challenge AC-state in the many practical settings. Instead, since our paper is primarily theoretical, we use experiments to highlight a conceptual limitation of AC-state compared to MISL.
>
> AC-state methods learn representations by predicting the first action taken between two states in a dataset. As a result, whether the learned representation is noise-invariant depends strongly on how that dataset was collected. In particular, if the data is generated by a noise-variant policy, the learned representation will also become sensitive to noise. Thus, in practice, these methods require access to a high-quality dataset collected by an appropriate expert or behavior policy.
>
> To test this directly, we make the data-collection policy imperfect by defining the 2D action as a linear combination of a state-dependent part and a noise-dependent part
> $w a_{\text{goal}}(x,y) + (1-w)\tanh(n)\vec{u}$.
>
> Where the goal is at $(10,10)$, $n$ is a scalar noise variable,$\vec{u}$ is a fixed 2D vector, and $w$ controls the weight on the state-based action. As we decrease $w$, the variance of the learned representation along the noise dimensions increases, for instace for $w=60%$ the representations are roughly as sensitive to noise as they are to the state dimensions, showing that AC-state is fundamentally sensitive to the dataset it is trained on.  Refer to [this Figure](https://anonymous.4open.science/r/MISL_representations-20ED/Picture1.pdf).
>
> In contrast, MISL representation learning does not rely on pre-collected data. It jointly learns the policy and the representation, and in doing so naturally induces a noise-invariant training policy.
>
> > Role of policy and discriminator architecture and size in minimality
>
> In our codebase, both the discriminator and policy are MLPs with 2 hidden layers of width 1024. We use no explicit regularization and still observe noise invariance in both the learned representations and the policy. To test the role of network expressivity, we also use a larger discriminator (\(n=8\), width \(2048\)) and a larger policy (\(n=5\), width \(1536\)), and separately increase the latent dimension from 2 to 16. Across all of these changes, both the policy and the representation remain noise-invariant. Increasing the latent dimension appears to slow the emergence of invariance slightly, but not substantially. Refer to [this Figure](https://anonymous.4open.science/r/MISL_representations-20ED/Figure%2014.pdf).
>
>
> > Pixel-based distractor environment
>
> Please refer to the **High-dimensional and pixel-based environments experiments** in the response to reviewer _PLSU_.
>
> ## Identifiability in stochastic MDPs Experiment
>
> We modify the point-maze environment so that each action in the $x$ or $y$ direction is applied only with probability $\alpha$; otherwise, no movement occurs along that direction. We then measure the coefficient of determination, $R^2$, of a linear probe trained to recover the underlying state from the learned representation. This is a standard way to test whether the representation preserves the relevant state information. For $\alpha \in \{0.2, 0.5, 0.7\}$, we obtain $R^2$ values close to 1 in fewer than $2 \times 10^5$ steps, showing that MISL still identifies the control-relevant features $(x,y)$ even in stochastic environments. Refer to [this Figure](https://anonymous.4open.science/r/MISL_representations-20ED/Figure%209.pdf).
>
> ## Action-interface invariance Experiment
>
> To test this, we modify the point-maze environment so that at the start of each episode the agent is placed in either room 0 or room 1 with probability $\frac{1}{2}$, and the state is augmented as $(x,y,\mathrm{roomID})$. In room 0, actions behave normally. In room 1, actions are rescaled by a factor of 2 while their range is reduced by half, so that the set of reachable future states remains unchanged. As a result, $(x,y,0)$ and $(x,y,1)$ for any $x,y$ have the same future possibilities despite different action parameterizations. We then compare the learned representations of these paired states and find that their normalized $\ell_2$ distance remains very low (less than $10^{-3}$) after $10^5$ environment interactions, showing that the representation is invariant to this change in action labeling. Refer to [this Figure](https://anonymous.4open.science/r/MISL_representations-20ED/Figure%2010.pdf).
>
> [1] Guaranteed Discovery of Control-Endogenous Latent States with Multi-Step Inverse Models, Lamb et al.

---

> > ### Author Rebuttal · Reviewer_y1ac · 2026-04-03
> >
> > Keep positive score

---

### Official Review · Reviewer_7AK9 · 2026-03-10

**Soundness:** 3
**Presentation:** 3
**Significance:** 3
**Originality:** 3
**Overall Recommendation:** 5
**Confidence:** 3

**Summary:**

This paper presents a theoretical study of the empowerment objective used in training reinforcement learning (RL) agents, with a particular focus on when empowerment is able to recover only control-relevant features of state. The authors frame empowerment as relying on "forward" and "backward" representations of states, which they use to derive assumptions under which empowerment ignores control-irrelevant features, as well as under which empowerment recovers all control-relevant features. The theoretical results are validated in an 8-state tabular environment and in the Point Maze environment.

**Compliance With Llm Reviewing Policy:**

Affirmed.

**Final Justification:**

My final score is 5.

My initial concerns were primarily the connection between theory and experiment, relation to bisimulation (and the wider literature), and the simple environments used in experiments.

The authors sufficiently addressed my concerns, adding a new task with image observations, and additional results making the connection to bisimulation more explicit. Assuming the authors integrate feedback from our discussion, as well as their discussions with over reviewers, particularly clarifying the scope of the paper and the relation between their method and other distractor-robust approaches, I believe this paper is solid and can be of use to the community, which can decide its impact.

Of course, further comparisons, both theoretical and empirical, to other distractor-robust approaches would be helpful. However, I do not believe that this is necessary to give this paper a positive score.

**Key Questions For Authors:**

1. The use of "induce" in the abstract seems overly strong. Does any empowerment agent induce forward and backward representations, or is that a parametrization assumption that you make in order to prove theoretical results?

2. You refer to "MISL policies", e.g., on l. 217. This is ambiguous if $p^\ast(z|s_0)$ is not a delta distribution. Are you implicitly assuming that $p^\ast$ has all mass concentrated on a single $z$ (justifiable by Lemma B.2?), and that $z$ gives a "MISL policy" $\pi_z$? If so, it should be clearly stated.

3. Is the discussion in the "Geometric intuition" paragraph under the assumptions of Theorem 5.1? If so, it should be clearly stated.

4. How do you define "irreducible noise process" in Corollary 5.6?

5. Regarding experiments: in Section 6.1, when you say "identical representations", is the $\ell_1$ distance between representations exactly $0$, or is there some numerical error? Also, Point Maze is not cited or defined. Do you use continuous actions?

**Limitations:**

Yes.

**Strengths And Weaknesses:**

**Strengths:**
- The paper tackles a well-motivated problem that is of interest to the RL community.
- The paper is mostly clearly written.
- The theoretical results look sound (although I did not look carefully through the proofs) and are support by the experimental results.
- The results are original to the best of my knowledge. I am familiar with some, but not all, related works, hence my confidence score of 3.

**Weaknesses (detailed later):**
- The theoretical claims could be tied into a larger context better. What is the upshot for applied researchers? Do the theoretical results provide any practical suggestions for designing empowerment-based objectives, or is this work intended as a theoretical justification of existing empirically successful methods?
- Bisimulation is mentioned and compared to multiple times throughout the paper, but is not defined. A brief definition would make the comparison cleaner and strengthen the theoretical claims.
- Some of the claims contain minor formatting errors.
- The environments used in experiments are simple. However, I appreciate that the aim of this work is primarily theoretical, and I believe it presents a relevant contribution to the community, which can judge its impact.

**Minor comments:**
- Typos, e.g., "has" on l. 85, "the all the" on l. 244-245, "Appendix G" on l. 352, $\ell 1$ instead of $\ell_1$, "symmetry" in l. 368, "trained in with" on l. 429.
- Confusing grammar in l. 42-49, right column.
- On l. 120, $s_n$ should be $s^+$?
- l. 176-180, left column, should be placed by Definition 4.1?
- What is $\mathcal{A}(s)$ in Theorem 5.1?

---

> ### Author Rebuttal · Authors · 2026-03-31
>
> > The theoretical claims tied into a larger context better.
>
> We believe this work helps position empowerment as a principled representation learning method with theoretical guarantees that shifts the focus of empowerment from exploration to representation learning. Our initial empirical results support the usefulness of these representations in practice. We view this work as laying the foundation for future empirical research on the usefulness of empowerment-based representations in more practical settings.
>
> > Bisimulation
>
> Our main reference is [1], which instantiated bisimulation by training a representation $\phi(s)$ with an objective of the form
> $J_{\phi}(\pi)=|\|\phi(s_i)-\phi(s_j)\|_1-|r_i-r_j|-\gamma W_2\left(\hat P(\cdot\mid \phi(s_i),a_i),\hat P(\cdot\mid \phi(s_j),a_j)\right)|,$
>
> ## Bisimulation baseline experiment.
> We highlight a key limitation of bisimulation. The goal of this experiment is not to show that MISL is always better than bisimulation; instead, the goal is to show the fundamental flaws of bisimulation that MISL is safe against, empirically. Bisimulation is expected to produce noise-invariant representations [1], but surprisingly, we find that it is sensitive to **temporally correlated noise** in practice. This is because bisimulation secretly relies on a learned transition model $\hat P(\cdot \mid \phi(s), a)$. When noise is temporally correlated, the model can predict future noise from current noise, causing states that differ only in noise to induce different next-state distributions and have different representations.
> In our experiment, we augment a point-maze with 5D Gaussian noise. With **temporally uncorrelated noise** and a dense goal-distance reward, the noise to state variance of the representations is at most $10^{-2}$ [Left Figure](https://anonymous.4open.science/r/MISL_representations-20ED/Picture2.pdf).
> Then we make the noise **strongly temporally correlated** (the noise simply increases by one in every timestep), and we observe that the variance along noise dimensions rises (exceeding $10^4$), showing the representations capture the noise [Right Figure](https://anonymous.4open.science/r/MISL_representations-20ED/Picture2.pdf). In contrast, MISL representations remain invariant to noise, no matter what the temporal structure of the noise is (Figure 5 of the paper).
>
>
> > environments are simple.
>
> Please refer to the response of reviewer _PLSU_ to see **our experiments in high-dimensional and pixel-based environments**.
>
> > Does any empowerment agent induce forward and backward representations?
>
> Optimizing empowerment through a variational lower bound naturally gives rise to forward and backward representations. Specifically, the variational formulation introduces a discriminator $q(z \mid s_0, s^+)$ that predicts the skill $z$ from the initial state $s_0$ and future state $s^+$. By forward and backward representations, we simply mean the information extracted from $s_0$ and $s^+$ for this discriminator. Thus, whenever empowerment is optimized via a variational lower-bound objective, such representations are learned, either explicitly or implicitly.
>
> > Are you implicitly assuming that $p^*(z \mid s)$ has all mass concentrated on a single $z$?
>
> No, we do not assume that $p(z \mid s)$ is a delta distribution. By definition, $p(z \mid s)$ is the optimizer of Eq. (2), that is, the optimal source distribution for channel capacity. Intuitively, it places mass on skills whose future outcomes are as distinct as possible, and therefore it will typically assign positive mass to multiple skills rather than collapsing to a single one, we refer to the set of the policies that have a mass under $p(z \mid s)$ as MISL policies. For a geometric intuition of this optimization problem and $p(z \mid s)$, we point the reviewer to Eysenbach et al. (2021), *The Information Geometry of Unsupervised Reinforcement Learning*.
>
> > discussion in the "Geometric intuition" and the assumptions of Theorem 5.1?
>
> No. The geometric intuition does not rely on any assumption. Its purpose is only to illustrate how the future-state polytope changes depending on whether both state features are control-relevant or whether one feature is control-irrelevant. No assumptions on the policy or representation are needed for this discussion.
>
> > How do you define "irreducible noise process"?
>
> By an irreducible noise process, we mean a process under which every state is reachable from every other state with nonzero probability.
>
> > In Section 6.1, when you say "identical representations"...
>
> Yes, by "identical representations" we mean representations whose $\ell_1$ distance is zero.
>
> > Point Maze
>
> Our Point Maze environment is a simple Gym-based continuous 2D navigation task developed by us. The state consists of $(x,y)$, and the action consists of $(a_x,a_y)$.
>
> [1] Learning Invariant Representations for Reinforcement Learning without Reconstruction. Amy Zhang, Rowan McAllister, Roberto Calandra, Yarin Gal, Sergey Levine

---

> > ### Author Rebuttal · Reviewer_7AK9 · 2026-04-02
> >
> > Thank you for your response. I trust that the authors will detail their Point Maze environment in the revised version of the manuscript, as this is an existing Gymnasium environment. Two remaining questions:
> > 1. The forward-backward factorization in Eq. 6 does not hold in general, for any $q$ and any representation dimension $d$ (unless, of course, the state space $\mathcal{S}$ has dimension $d$, but I believe that isn't stated in the text). E.g., if $q$ is a function of $s_0^\top s^+$ but $\phi$ and $\psi$ map a high-dimensional $\mathcal{S}$ to a low-dimensional $\mathbb{R}^d$. In general, I assume we want $\phi$ and $\psi$ to reduce the dimensionality of states. Can the authors comment on whether this is true? Should Eq. 6 be stated as an explicit assumption, rather than a given? What dimensionalities did you use in practice for your experiments?
> > 2. The Ant and Kitchen experiments you provide are nice. Echoing Reviewer S9NU, are you able to provide policy-learning results for Kitchen? Also, do you have policy-learning results (on any task) of MISL vs. bisimulation, rather than MISL vs. raw state?

---

> > > ### Author Response · Authors · 2026-04-04
> > >
> > > We thank the reviewer for the acknowledgement and for the constructive comments.
> > >
> > > > pointMaze
> > >
> > > The **PointMaze** environment we use is based on the **Gymnasium PointMaze** (a minimal version of it that we implemented), but there are three differences:
> > > 1. We do **not** include the velocity or the desired goal location as part of the state.
> > > 2. Our goal-reaching reward function is slightly different: we use the **negative Euclidean distance** to the goal as the dense reward, and when the Euclidean distance is less than **1**, we assign a reward of **10**.
> > > 3. We augment the observations with an additional **noise process**.
> > >
> > > We will certainly make sure to clarify these differences in the final version of the paper.
> > >
> > > > The forward-backward factorization
> > >
> > > The required dimensionality of the representations can indeed depend on how complicated the ground-truth posterior is. However, when we define these representations (eq 6), **we do not make any assumption about their dimensionality**. In general, they can have the same dimensionality as the state observations. We already tried to make this point somewhat clear around **line 133 (right)**, namely that the definition itself does not preclude representing any posterior, since the representation dimension can in principle be as high as needed. But I agree with the reviewer that this should be stated more explicitly so that it does not appear to be an assumption.
> > >
> > > That said, the main contribution of our paper is that we theoretically and empirically characterize the situations in which **MISL representations can be much lower-dimensional than the full state**. For instance, Theorem 5.1 shows **invariance to action relabeling**: if there are states that have similar available actions up to a permutation, then those states will have similar forward representations, so less expressivity is needed. Similarly, Theorem 5.4 and Corollary 5.6 show that **in the presence of control-irrelevant features, the representations become invariant to them**, again implying a **lower-dimensional representation** than the full state space.
> > >
> > > For example, if the state is 5-dimensional, but 2 dimensions are control-irrelevant, and 1 additional dimension only distinguishes states with the same future outcomes but relabeled actions, then the minimal MISL representations would only need to be 2-dimensional. We will make sure this point is stated much more clearly in the final version of the paper.
> > >
> > > > What dimensionalities did you use in practice for your experiments?
> > >
> > > In our experiments, the learned representations are indeed **much lower-dimensional than the full observation space** while still working well in practice. For example, in **PointMaze** we used a representation dimension of **\(d=2\)**, while the actual observation is **7-dimensional** (**2** state dimensions + **5** noise dimensions). In **Ant**, we used **\(d=10\)**, while the full state is **39-dimensional** (**29** state dimensions + **10** noise dimensions). In **Kitchen**, we used **\(d=24\)**, while the observation space is much higher-dimensional, consisting of **\(64 \times 64 \times 3\)** image observations together with an additional **\(10 \times 10 \times 3\)** noise patch. Thus, across all of our experiments, the learned representations are **substantially lower-dimensional than the full state/observation space**, yet still work well in practice.
> > >
> > > > Kitchen policy training
> > >
> > > Please check the results of this experiment in our rebuttal reply to reviewer S9NU. Thanks for your attention!
> > >
> > > > Bisimulation policy training
> > >
> > > We believe that comparing policy training with bisimulation representations directly against MISL is not fully fair.
> > > Bisimulation representations are trained for a specific reward function, so they are naturally **specialized** to that reward. This **assumes that the reward is available during representation learning** and that this is the only reward we care about. As a result, bisimulation representations may **not generalize well to other reward functions**.
> > > In contrast, MISL is fully unsupervised, and **does not require access to the reward** during representation learning. It is designed to capture all control-relevant features, making it **useful for a broad range of downstream reward functions**.
> > > We therefore believe a fairer comparison is the following: train bisimulation representations on one reward function, freeze them, and then use them to train a policy and critic on a slightly different reward function.
> > > We performed this experiment in the noisy PointMaze. We trained bisimulation representations using a goal at location `(2,2)`, then froze the representations and trained the policy for a new goal at `(10,10)`. We compared this with policy training using MISL representations and found comparable results [Figure](https://anonymous.4open.science/r/MISL_representations-20ED/misl_vs_bisim_episode_return.pdf).

---

### Official Review · Reviewer_PLSU · 2026-03-13

**Soundness:** 2
**Presentation:** 3
**Significance:** 3
**Originality:** 3
**Overall Recommendation:** 4
**Confidence:** 3

**Summary:**

Within the context of reinforcement learning and skill learning, the paper focuses on using an empowerment objective which maximizes an agent's influence over a domain to theoretically and empirically explore if only control-relevant features are capturable. Further the paper introduces MISL forward and backward representations as a conceptual tool and explores the conditions where MISL features are all control-irrelevant.

**Compliance With Llm Reviewing Policy:**

Affirmed.

**Final Justification:**

The rebuttal addressed my questions - I will keep my positive overall recommendation.

**Key Questions For Authors:**

- For (Figure 6) and RQ5 (Identifiability of control-relevant features) in the appendix, is it showing that MISL only captures control relevant features or that it's capturing all of the control relevant features?

- The evaluation side is quite limited in terms of domains with more complex dynamics given the controllability context, why has further evaluations such as those in 'complex control or pixel-based environments'
not been included?

- A recent paper has introduced plasticity as the mirror of empowerment (https://openreview.net/forum?id=eOZFqyE9Ok), does the theoretical contributions of this paper extend to the notion of plasticity?

**Limitations:**

yes

**Strengths And Weaknesses:**

Strengths:

- The authors provide compelling theoretical results on the how MISL policies are invariant to control-irrelevant features of the environment.
- The paper is generally well written and well presented with appropriate detail provided in the background and appendix.
- I believe these results will be useful for further research and use cases of empowerment.

Weaknesses:

- There is only a limited amount of empirical evaluation which makes it difficult to know how well the theoretical results extend in practice.

---

> ### Author Rebuttal · Authors · 2026-03-31
>
> Thanks to the reviewer for the thoughtful and helpful comments. Below, we address the concerns and questions raised.
>
> > Limited amount of empirical evaluation
>
> We first emphasize that the main contribution of this paper is theoretical. Our goal is to formally characterize the forward and backward representations learned through empowerment-based objectives, including their properties, invariances, and how they differ from each other. That said, to better support the theory with empirical evidence, we have added the following new experiments:
>
> ## High-dimensional and pixel-based environment experiments
> We evaluate noise invariance in both **MuJoCo Ant** and the **pixel-based Lexa Kitchen** environment. In Ant, we augment the 29-dimensional continuous state with 10 i.i.d. Gaussian noise dimensions with std=20 and compare the covariance of the learned representation when only the noise dimensions vary versus when only the true state dimensions vary. The resulting noise-to-state variance ratio falls below $10^{-4}$ after around 1M environment steps and continues to decrease ([Left Figure](https://anonymous.4open.science/r/MISL_representations-20ED/Figure%2011.pdf)). These representations are also useful for downstream control: in the same noisy environment, we train the ant with the run forward reward (a combination of x velocity and staying healthy), training with raw states gives returns below 300, whereas using MISL representations consistently reaches returns above 800 within 1M steps [Figure](https://anonymous.4open.science/r/MISL_representations-20ED/Figure%2012.pdf).
>
> In the kitchen, we add a $10 \times 10 \times 3$ patch of uniform noise (uniform to match the scale of the normalized pixels) to the $64 \times 64 \times 3$ RGB observation and perform the same covariance-based analysis. There as well, the noise-to-state variance ratio of the representations drops below $10^{-3}$ after 1M interaction steps and keeps decreasing ([Right Figure](https://anonymous.4open.science/r/MISL_representations-20ED/Figure%2011.pdf). These experiments verify the noise invariance of MISL representations in high dimensional enviornments and their usefulness for downstream tasks in such environments.
>
>
> In addition, we would like to draw the reviewer’s attention to the **new experiments comparing our method with bisimulation-based baselines and AC-State methods**, which are included in the rebuttal to the reviewer _7AK9, y1ac_ , respectively. We also included additional experiments verifying the **practical validity of Theorem 5.1** and the **identifiability of the learned representations in stochastic environments** in response to reviewer _y1ac_. which further strengthens our empirical results.
>
>
> > For Figure 6 and RQ5 (Identifiability of control-relevant features) in the appendix, is it showing that MISL only captures control-relevant features, or that it captures all control-relevant features?
>
> These two experiments address different aspects:
>
> - **Figure 6** shows that MISL representations are **invariant to unseen noise profiles** at test time. In this experiment, we train MISL representations with temporally independent noise, then evaluate their sensitivity to noise versus state dimensions under temporally correlated noise. We find that the variance due to noise remains much lower, supporting noise invariance.
>
> - **RQ5**, on the other hand, addresses whether MISL captures **all control-relevant features**. Specifically, we compute the L1 distance between representations for all pairs of endogenous states and confirm that it is nonzero for every pair. This indicates that no two states with different control-relevant features are mapped to the same representation.

---

> > ### Author Rebuttal · Reviewer_PLSU · 2026-04-03
> >
> > I thank the authors for the response. I will keep my positive score.

---

### Decision · Program_Chairs · 2026-04-30

**Decision:**

Accept (regular)

**Comment:**

All reviewers agree that the paper provides an interesting and novel approach for identifying control-relevant features through a newly proposed empowerment objective, as well as being well-written and well-presented. Most of the initial concerns of the reviewers have been addressed in the exhaustive and well-argued rebuttal, although further comparisons to other methods that can deal with distractors would strengthen the paper even further. Overall, this is a very solid paper and I'm sure addressing also the other reviewers' suggestions will further improve its final version.